# Tree ensemble kernels for Bayesian optimization with known constraints over mixed-feature spaces

**Alexander Thebelt** [*]
Imperial College London
London, UK

**Calvin Tsay**
Imperial College London
London, UK

**Robert M. Lee**
BASF SE
Ludwigshafen, Germany

**Nathan Sudermann-Merx**
Cooperative State University
Mannheim, Germany

**David Walz**
BASF SE
Ludwigshafen, Germany

**Behrang Shafei**
BASF SE
Ludwigshafen, Germany

**Ruth Misener**
Imperial College London
London, UK

## Abstract

Tree ensembles can be well-suited for black-box optimization tasks such as algorithm tuning and neural architecture search, as they achieve good predictive performance with little or no manual tuning, naturally handle discrete feature spaces, and are relatively insensitive to outliers in the training data. Two well-known challenges in using tree ensembles for black-box optimization are (i) effectively quantifying model uncertainty for exploration and (ii) optimizing over the piece-wise constant acquisition function. To address both points simultaneously, we propose using the kernel interpretation of tree ensembles as a Gaussian Process prior to obtain model variance estimates, and we develop a compatible optimization formulation for the acquisition function. The latter further allows us to seamlessly integrate known constraints to improve sampling efficiency by considering domain-knowledge in engineering settings and modeling search space symmetries, e.g., hierarchical relationships in neural architecture search. Our framework performs as well as state-of-the-art methods for unconstrained black-box optimization over continuous/discrete features and outperforms competing methods for problems combining mixed-variable feature spaces and known input constraints.

## 1 Introduction

Many black-box optimization problems contain feature relationships known *a priori* based on domain knowledge, such as hierarchies or constraints [51, 68]. For example, hierarchical structures arise in neural architecture search [21, 72], where hyperparameters such as kernel size are only relevant if a convolutional layer is selected. Explicit constraints can also arise, e.g., matching kernel size and stride to input channel size and padding. In many cases, Bayesian optimization can incorporate known hierarchies and/or constraints, given that a suitable surrogate model is selected. To this end, Fromont et al. [23] and Nijssen and Fromont [53] impose a variable hierarchy by constraining the splitting order of decision trees, i.e., certain attributes must be selected before others.

Tree-based models, such as random forests or gradient-boosted trees, remain popular in many applications, as they inherit the innate ability of simple decision trees to seamlessly handle categorical and

---

[*]Corresponding author: alexander.thebelt18@imperial.ac.uk

36th Conference on Neural Information Processing Systems (NeurIPS 2022).

discrete input spaces. Moreover, they are highly parallelizable and scalable to high-dimensional data. Despite these modelling advantages, the deployment of tree-based models in Bayesian optimization has been limited by challenges pertaining to (i) quantifying prediction uncertainty and (ii) optimizing acquisition functions defined by their discontinuous response surfaces [61]. Early works, e.g., the popular SMAC algorithm [34], addressed (i) using empirical variance within a tree ensemble and (ii) via local and/or random search methods. Moreover, recent works [49, 50, 66] propose mixed-integer formulations for tree ensembles, enabling optimization over their mean functions.

**Contributions.** Sections 2 and 3 present related work and methods used to derive the approach proposed in this paper. We present a mixed-integer second-order cone optimization formulation for tree kernel Gaussian processes in Section 4 and show that that the *tree agreement ratio*, i.e., the hyperparameter introduced by the tree ensemble kernel, sufficiently represents the model uncertainty in Section 5.1. Section 5.2 shows that solving the mixed-integer second-order cone optimization problem considerably outperforms sampling-based strategies in Bayesian optimization. Our approach of using tree ensemble kernels as a Gaussian process prior is particularly useful for applications combining mixed-variable spaces and known input constraints. A Python implementation of the proposed algorithm is available at: `www.github.com/cog-imperial/tree_kernel_gp`

## 2 Related work

Bayesian optimization (BO) solves [22, 27, 61]: $\mathbf{x}_f^* \in \arg\max_{\mathbf{x}} f(\mathbf{x})$, where $f$ is an expensive-to-evaluate black-box function that can be queried at inputs $\mathbf{x} \in \mathcal{X}$ to derive the optimal solution $\mathbf{x}_f^*$. BO iteratively updates a surrogate model of $f$ and optimizes a corresponding acquisition function that balances exploitation and exploration. Maximizing the acquisition function produces a new query $\mathbf{x}^*$ which is evaluated and added to the set of observations. Gaussian processes (GPs) [57] are a common choice of BO surrogate due to their flexibility, e.g., domain-specific knowledge can be built into the GP prior via mean and kernel functions, and reliable uncertainty quantification to identify unexplored search areas. Open-source tools such as BoTorch [2] implement BO with GP surrogates and offer a wide selection of kernels mainly suited for continuous search spaces. Prior works developed GPs with modified kernels to integrate discrete features [10, 19, 25, 30, 59] or considered conditional feature spaces [29, 35, 43, 48]. Nguyen et al. [52] integrate catgorical and category-specific continuous inputs by formulating the black-box optimization problem as a multi-arm bandit problem for which each category corresponds to an arm. Similarly, Gopakumar et al. [26] handle mixed-type inputs using multi-armed bandits. Besides GPs, tree ensemble-based surrogates show excellent performance for black-box optimization with mixed-variable settings, i.e., with continuous, integer and categorical variables, and for structured search spaces, e.g., hierarchical and conditional feature spaces [34].

Black-box optimization tools using tree ensembles, e.g., `SMAC` [34] and Scikit-Optimize (`SKOPT`) [31], are useful for applications such as neural architecture search (NAS) and algorithm tuning. Shahriari et al. [61] mention challenges in deploying tree ensembles for BO: (i) quantifying uncertainty for exploration purposes, and (ii) optimizing over the non-differentiable discrete acquisition function to determine the next query point. `SMAC` identifies uncertain search space regions using empirical variables across tree predictions of the random forest and optimizes the acquisition function combining local and random search. Bergstra et al. [5] proposes the Tree Parzen Estimator (TPE) to handle categorical variables and conditional structures by modeling individual input dimensions by a kernel density estimator. However, the TPE approach ignores dependencies between dimensions. For gradient-boosted tree ensembles, `SKOPT` derives uncertainty with quantile regression to fit two models for the 16th and 84th percentile and averages the predictions to estimate the standard deviation. For random forests, `SKOPT` uses an uncertainty strategy similar to `SMAC`. In general, `SKOPT` relies on random sampling to optimize the acquisition function.

Mišić [49] proposed a mixed-integer optimization formulation for tree ensemble mean functions that has been used in several applications [11, 50, 65–67]. Besides improving the solution to an acquisition function, mixed-integer formulations also allow explicit consideration of input constraints to incorporate domain knowledge. Some software tools, e.g., BoTorch, support linear equality and inequality constraints of continuous variables at the acquisition function optimization step, while tree ensemble-based algorithms do not support input constraints. Papalexopoulos et al. [54] use ReLU neural networks as surrogate models and deploy a mixed-integer linear formulation to optimize the acquisition function. The approach relies on random initialization and stochasticity in the model training to allow for exploration. Daxberger et al. [16] handle mixed-variable search spaces by using

a Bayesian linear regressor that uses an integer solver to search the discrete subspace. The authors introduce features capturing the discrete parts of the search space by using a BOCS model [3, 18], while continuous parts are handled with random Fourier features [56]. Genetic Algorithms (GA) are another class of algorithms, which deploy evolution-based selection heuristics to maximize black-box functions [36]. While there is no feasibility guarantee for input constraints, GA implementations like pymoo [7] support constraint optimization by minimizing constraint violation.

We compare our BO approach, which uses the kernel interpretation of tree ensembles as a Gaussian process prior, to: SMAC, the random forest and gradient-boosted tree versions of SKOPT, (SKOPT-RF, and SKOPT-GBRT, respectively) in the Section 5 numerical studies as a baseline for other tree ensemble-based algorithms. We also compare against the default upper-confidence bound and expected improvement BO implementations of BoTorch (UCB-MATERN and EI-MATERN, respectively) and the default GA algorithm of pymoo to include black-box algorithms that partially support constrained optimization.

## 3 Technical background on prior work

### 3.1 Tree ensemble kernel as a Gaussian process prior

Exploring the search space requires quantifying the uncertainty of the underlying surrogate model. We use the kernel interpretation of tree ensembles based on random partitions [15, 73]. The tree kernel captures correlation between two input data points $(\mathbf{x}, \mathbf{x}') \in \mathbb{R}^n$:

$$k_{\text{Tree}}(\mathbf{x}, \mathbf{x}') = \sigma_0^2 \, |\mathcal{T}|^{-1} \, \mathbf{z}(\mathbf{x})^\intercal \mathbf{z}(\mathbf{x}') \tag{1}$$

To derive the tree kernel, we first train a gradient boosted tree ensemble $\mathcal{T}$ on data set $\mathbf{X} \in \mathbb{R}^{m \times n}$ with $n$ denoting the dimensionality of the search space and $m$ the size of the data set. Every tree $t$ in the tree ensemble $\mathcal{T}$ maps inputs $\mathbf{x} \in \mathbb{R}^n$ onto a leaf $l$ by sequentially evaluating splitting conditions. Each leaf $l$ defines a subspace $\mathbf{x}_l \subset \mathbb{R}^n$ restricted by active splits $s \in \mathbf{splits}(t)$. Two inputs $(\mathbf{x}, \mathbf{x}')$ are fully correlated in tree $t$ if both end up in the leaf subspace $\mathbf{x}_{t,l}$ and uncorrelated otherwise. The Eq. (1) vector $\mathbf{z}(\mathbf{x})$ consists of binary elements $z_{t,l}$ indicating if leaf $l \in \mathcal{L}_t$ is active for input $\mathbf{x}$, with $\mathcal{L}_t$ denoting the set of all leaves in tree $t$. The inner product $\mathbf{z}(\mathbf{x})^\intercal \mathbf{z}(\mathbf{x}')$, normalized by the total number of trees $|\mathcal{T}|$, gives the ratio of trees in the ensemble for which $\mathbf{x}$ and $\mathbf{x}'$ fall into the same leaf. We modify the kernel by adding a trainable signal variance $\sigma_0^2$ [42]. Note that the tree ensemble defining the kernel and the kernel hyperparameters are trained separately. Davies and Ghahramani [15] prove that the tree kernel is a suitable GP prior. The resulting (non-stationary and supervised) tree kernel describes a prior over piece-wise constant functions when used in a GP.

### 3.2 Posterior distribution

We approximate $f$ as a Gaussian process with zero mean and kernel $k_{\text{Tree}}$: $f(\cdot) \sim \mathcal{GP}(0, k_{\text{Tree}})$. Since $k_{\text{Tree}}$ is a valid Mercer kernel, the mean $M(\mathbf{x})$ and variance $V(\mathbf{x})$ of the GP at $x \in \mathbb{R}^n$ is [57]:

$$M(\mathbf{x}) = K_{\mathbf{x},\mathbf{X}} \, (K_{\mathbf{X},\mathbf{X}})^{-1} \, \mathbf{y} \tag{2a}$$

$$V(\mathbf{x}) = K_{\mathbf{x},\mathbf{x}} - K_{\mathbf{x},\mathbf{X}} \, (K_{\mathbf{X},\mathbf{X}})^{-1} \, K_{\mathbf{x},\mathbf{X}}^\intercal \tag{2b}$$

The Gram matrix $K_{\mathbf{X},\mathbf{X}} \in \mathbb{R}^{m \times m}$ has entries describing pairwise correlations computed based on the kernel function in Eq. (1) . The entries of vector $K_{\mathbf{x},\mathbf{X}} \in \mathbb{R}^m$ contain correlations between the input $\mathbf{x}$ and sampled data points, defined as $[k_{\text{Tree}}(\mathbf{x}, \mathbf{x}_1), k_{\text{Tree}}(\mathbf{x}, \mathbf{x}_2), \ldots, k_{\text{Tree}}(\mathbf{x}, \mathbf{x}_m)]$ with vectors $\mathbf{x}_i$ referring to rows in data set $\mathbf{X}$. Target values $\mathbf{y} \in \mathbb{R}^m$ are the corresponding observations for $\mathbf{X}$. Eq. (2) describes the noise-free case of the GP mean and variance. To fit a GP based on the tree kernel function, we usually require a noise term, i.e., a diagonal matrix $\sigma_y^2 I$ that is added to $K_{\mathbf{X},\mathbf{X}}$. We set hyperparameters $\sigma_y^2$ and $\sigma_0^2$ by maximizing the log marginal likelihood. Lee et al. [42] compute the Eq. (2a) inverse of the Gram matrix $K_{\mathbf{X},\mathbf{X}}$ efficiently by exploiting the property that the rank of $K_{\mathbf{X},\mathbf{X}}$ is at most the number of leaves over trees. Fig. 1 visualizes the Eq. (2) $M(\mathbf{x})$, $V(\mathbf{x})$ and the upper-confidence bound (UCB) [14] response surface. Fig. 1(a) shows that the tree kernel-based GP mean gives a good piecewise-constant approximation of negative one times the Branin function. Fig. 1(b) shows that variance peaks reveal areas where data are sparse, reliably identifying uncertainty in the underlying surrogate model. Finally, Fig. 1(c) shows how an acquisition function such as the UCB can effectively manage the exploitation-exploration trade-off.

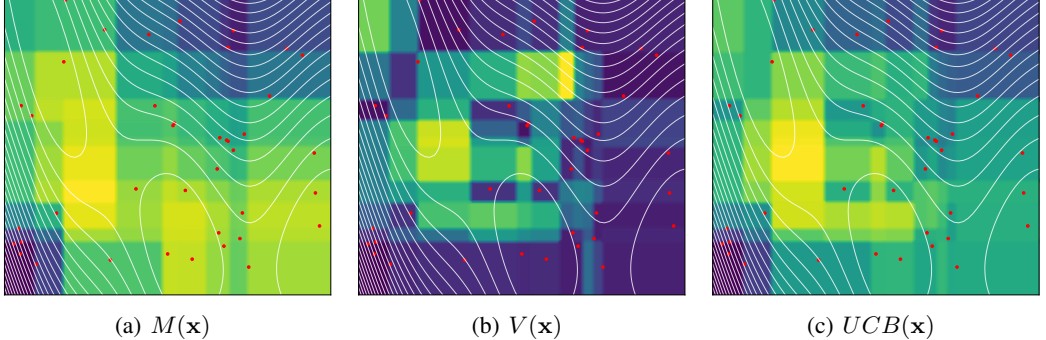

(a) $M(\mathbf{x})$        (b) $V(\mathbf{x})$        (c) $UCB(\mathbf{x})$

Figure 1: Tree kernel GP trained on 40 random points of negative one times the Branin function for the intervals $\mathbf{x} \in ([-5.0, 10.0], [0.0, 15.0])^{\mathsf{T}}$. Function values increase with the colour brightness and white contour lines indicate the true functional shape of the negated Branin function.

## 3.3 Global optimization of tree ensembles

Mišić [49] proposes a mixed-integer linear optimization formulation that ensures that binary variables $z_{t,l}$ follow the logic of the tree ensemble:

$$\sum_{l \in \mathcal{L}_t} z_{t,l} = 1, \qquad\qquad\qquad \forall t \in \mathcal{T}, \quad (3a)$$

$$\sum_{l \in \mathbf{left}(s)} z_{t,l} \leq \sum_{j \in \mathbf{C}(s)} \nu_{\mathrm{V}(s),j}, \qquad\qquad\qquad \forall t \in \mathcal{T}, \forall s \in \mathbf{splits}(t), \quad (3b)$$

$$\sum_{l \in \mathbf{right}(s)} z_{t,l} \leq 1 - \sum_{j \in \mathbf{C}(s)} \nu_{\mathrm{V}(s),j}, \qquad\qquad\qquad \forall t \in \mathcal{T}, \forall s \in \mathbf{splits}(t), \quad (3c)$$

$$\sum_{j=1}^{K_i} \nu_{i,j} = 1, \qquad\qquad\qquad \forall i \in \mathcal{C}, \quad (3d)$$

$$\nu_{i,j} \leq \nu_{i,j+1}, \qquad\qquad\qquad \forall i \in \mathcal{N}, \forall j \in [K_i - 1], \quad (3e)$$

$$\nu_{i,j} \in \{0,1\}, \qquad\qquad\qquad \forall i \in [n], \forall j \in [K_i], \quad (3f)$$

$$z_{t,l} \geq 0, \qquad\qquad\qquad \forall t \in \mathcal{T}, \forall l \in \mathcal{L}_t. \quad (3g)$$

Eq. (3a) guarantees exactly one active leaf $l$ in leaf set $\mathcal{L}_t$ for tree $t$. Eqs. (3b)–(3c) ensure that binary variables $z_{t,l}$ are only active if all previous split binaries $\nu_{\mathrm{V}(s),j}$ corresponding to continuous splitting thresholds are active. At any node in a given tree, $\mathbf{left}(s)$ and $\mathbf{right}(s)$ contain all leaves following the left and right branches, respectively. The mapping $\mathrm{V}(s)$ gives the splitting feature at node $s \in \mathbf{splits}(t)$ in tree $t$, with $\mathbf{splits}(t)$ defining the set of all splits in tree $t$. Tree ensembles can handle continuous, integer and categorical data. Continuous splits $s$ are defined by $x_{\mathrm{V}(s)} \leq \upsilon_{\mathrm{V}(s),j}$ conditions, where $\upsilon_{\mathrm{V}(s),j}$ is the learned splitting threshold. Therefore, $\mathbf{C}(s)$ only contains a single index $j$ representing the threshold $\upsilon_{\mathrm{V}(s),j}$. Categorical splits $s$ are characterized by (subsets of) categories available for feature $\mathrm{V}(s)$ and define a splitting condition based on the inclusion of $x_{\mathrm{V}(s)}$ in the category subset at split $s$. For categorical splits, $\mathbf{C}(s)$ includes the category indices comprising the category subset at split $s$. Eq. (3d) ensures that only one category is active per categorical variable $i \in \mathcal{C}$. Continuous splitting thresholds of all trees in the ensemble are ordered according to $v_{i,1} < v_{i,2} < ... < v_{i,K_i}$ with $K_i$ denoting the index for the last split of continuous feature $i \in \mathcal{N}$. To enforce this order, Eq. (3e) ensures that binary variables $\nu_{i,j}$, corresponding to the split thresholds $v_{i,j}$, are activated sequentially. The model comprising Eqs. (3) has no direct dependency on $\mathbf{x}$ and is fully defined by binary variables indicating which splits and leaves of the tree model active. However, to allow the user to include extra equality and inequality constraints on the input vector $\mathbf{x}$, we bound

the continuous variables based on the active splits by adding linking constraints [50]:

$$x_i \geq v_i^L + \sum_{j=1}^{K_i} (v_{i,j} - v_{i,j-1})(1 - \nu_{i,j}), \qquad \forall i \in \mathcal{N}, \quad (4a)$$

$$x_i \leq v_i^U + \sum_{j=1}^{K_i} (v_{i,j} - v_{i,j+1})\nu_{i,j}, \qquad \forall i \in \mathcal{N}, \quad (4b)$$

$$x_i \in [v_i^L, v_i^U], \qquad \forall i \in \mathcal{N}, \quad (4c)$$

$$x_i = \{j \in [K_i] \mid \nu_{i,j} = 1\}, \qquad \forall i \in \mathcal{C}, \quad (4d)$$

For categorical variables, Eq. (4d) maps the indices of active categories onto $x_i$ for $i \in \mathcal{C}$. These optimization formulations are implemented in open-source software ENTMOOT [66] and OMLT [11].

## 4    Tree ensemble kernels for Bayesian optimization

The technical details in Section 3 are insufficient to use the tree ensemble kernel in a Bayesian optimization framework. While Eqs. 3 and 4 allow optimization over the mean of an associated acquisition function [49], Bayesian optimization also requires quantifying model uncertainty for exploration. This section proposes a mixed-integer second-order cone optimization formulation to capture the standard deviation of a GP with a tree kernel prior. Combining this optimization formulation with the already-developed mixed-integer formulation of the mean function, we derive the upper-confidence bound (UCB) of the tree kernel-based GP. The advantage of deriving a mixed-integer second-order cone optimization formulation of the UCB acquisition function is that we can globally optimize the acquisition function. Additionally, we can easily incorporate explicit input constraints that capture domain knowledge and/or known search space relationships.

Our optimization problem, which includes the UCB acquisition function, is:

$$\mathbf{x}_{\text{lb}}^*, \mathbf{x}_{\text{ub}}^*, \mathbf{x}_{\text{cat}}^* \in \arg\max_{\mathbf{x}} \mu(\mathbf{x}) + \kappa\sigma(\mathbf{x}), \qquad (5a)$$

$$h(\mathbf{x}) = 0, \qquad (5b)$$

$$g(\mathbf{x}) \leq 0, \qquad (5c)$$

with $\mu(\mathbf{x})$ and $\sigma(\mathbf{x})$ denoting the surrogate model's mean prediction and standard deviation, respectively. Functions $h(\mathbf{x})$ and $g(\mathbf{x})$ are known constraints and handled similarly to [9, 69]. In our implementation, the constraints can be linear, quadratic, or polynomial. Hyperparameter $\kappa \geq 0$ controls the exploitation-exploration trade-off to determine the next black-box function query area. The solution to Eq. 5a is defined by $\mathbf{x}_{\text{lb}}^*$ and $\mathbf{x}_{\text{ub}}^*$ for non-categorical variables $i \in \mathcal{N}$, i.e., continuous and integer features, and $\mathbf{x}_{\text{cat}}^*$, a set of valid category subsets for categorical variables $i \in \mathcal{C}$. Two vectors ($\mathbf{x}_{\text{lb}}^*$ and $\mathbf{x}_{\text{ub}}^*$) define the non-categorical variables because the trees are piecewise-constant over intervals and the vectors define the lower and upper bounds of these intervals.

Next, we formalize the mean and variance of the tree kernel-based GP. Eqs. 6 and 2 are equivalent.

$$\mu(\mathbf{x}) = M(\mathbf{x}) = K_{\mathbf{x},\mathbf{X}}(K_{\mathbf{X},\mathbf{X}})^{-1}\mathbf{y} \qquad (6a)$$

$$\sigma^2(\mathbf{x}) = V(\mathbf{x}) = K_{\mathbf{x},\mathbf{x}} - K_{\mathbf{x},\mathbf{X}}(K_{\mathbf{X},\mathbf{X}})^{-1}K_{\mathbf{x},\mathbf{X}}^{\mathsf{T}} \qquad (6b)$$

The Gram matrix $K_{\mathbf{X},\mathbf{X}}$ and target vector $\mathbf{y}$ are constants in the optimization model, as these quantities only depend on the data set $\mathbf{X}$. The value $K_{\mathbf{x},\mathbf{x}}$ is directly related to signal variance hyperparameter $\sigma_0$ since there is full leaf overlap for two identical inputs:

$$K_{\mathbf{x},\mathbf{x}} = \sigma_0^2 \qquad (7)$$

The vector $K_{\mathbf{x},\mathbf{X}}$ contains the kernel output of $\mathbf{x}$ with individual data points $\mathbf{x}_i$. We compute the constant matrix $A \in \mathbb{R}^{m \times |\mathcal{L}|}$ after the gradient-boosted tree is trained but before solving the acquisition function. The entries of $A$ are equal to 1 for all active leaves $l$ of data point $i$ and 0 otherwise. Given matrix $A$, Eqs. (8) capture the kernel output by summing over binary variables $z_{t,l}$ that are active for data point $i$.

$$K_{\mathbf{x},\mathbf{X}} = [k_{\text{Tree}}(\mathbf{x}, \mathbf{x}_1), k_{\text{Tree}}(\mathbf{x}, \mathbf{x}_2), \dots, k_{\text{Tree}}(\mathbf{x}, \mathbf{x}_m)] \qquad (8a)$$

$$k_{\text{Tree}}(\mathbf{x}, \mathbf{x}_i) = \sigma_0^2 \, |\mathcal{T}|^{-1} \sum_{t \in \mathcal{T}} \sum_{l \in \mathcal{L}_t} A_{i,l} z_{t,l} \qquad\qquad \forall i \in [m] \qquad (8b)$$

To get an intuition for Eqs. (8), note that the largest possible value for each $k_{\text{Tree}}(\mathbf{x}, \mathbf{x}_i)$ is $\sigma_0^2$ (if the corresponding entries of matrix $A$ are all equal to 1, and there is full leaf overlap) and the smallest possible value for each $k_{\text{Tree}}(\mathbf{x}, \mathbf{x}_i)$ is 0 (if there is no leaf overlap). In general, values of each element of vector $K_{\mathbf{x},\mathbf{X}}$ will range between 0 and $\sigma_0^2$: higher values in the elements of $K_{\mathbf{x},\mathbf{X}}$ indicate a higher degree of overlap between the next query location $\mathbf{x}$ and data point $\mathbf{x}_i$ in the set $\mathbf{X}$.

Without considering additional tree model constraints $h$ and $g$, the resulting acquisition function is a mixed-integer quadratic optimization problem. The optimization problem is *mixed-integer* because of binary variables $\boldsymbol{\nu}$ and *quadratic* because of Eq. (6b). The quadratic Eq. (6b) components are $\sigma^2$ and the terms $k_{\text{Tree}}^2(\mathbf{x}, \mathbf{x}_i)$ arising from the inner product of $K_{\mathbf{x},\mathbf{X}}$ with itself. We only require one direction of the Eq. (6b) equality ($\leq$) and re-write Eq. (6b) as a second-order cone constraint. Second-order cone programming [1, 41, 45] optimizes over a linear objective subject to both linear and second-order cone constraints (here, convex quadratic constraints, but the theory is more general). More recently, solvers integrate advanced methods solving second-order cone problems in the mixed-integer setting [4, 20, 46]. Solver Gurobi 9 [28] automatically finds that Eq. (6b) (with $\leq$ rather than $=$) can be represented as a second-order cone and makes the appropriate algorithm modifications, e.g., as described by [33, 71]. The proposed formulation is also compatible with open-source solver alternatives including Bonmin [8], MindtPy [6], Pajarito [13], and SHOT [47].

The acquisition function, with Objective 5a and Constraints (3)–(8b), is a mixed-integer second-order cone program which can be solved with optimization solvers. A valid solution to the proposed model is a set of active leaves $\mathcal{L}^*$ and the intersection of all leaf subspaces $[\mathbf{x}_{\text{lb}}^*, \mathbf{x}_{\text{ub}}^*], \mathbf{x}_{\text{cat}}^*$.

To derive the next black-box function query point $\mathbf{x}^*$, we propose some heuristics. The acquisition function value is constant for $\boldsymbol{x} \in [\mathbf{x}_{\text{lb}}^*, \mathbf{x}_{\text{ub}}^*], \mathbf{x}_{\text{cat}}^*$, i.e., all contained points are equivalent from the perspective of the tree kernel-GP. For the continuous and integer features, we observe that tree models tend to learn split thresholds close to training data points. We propose the center of $[\mathbf{x}_{\text{lb}}^*, \mathbf{x}_{\text{ub}}^*]$ as the next query point $\mathbf{x}^*$ for continuous and integer features:

$$x_{\text{mid},i}^* = \frac{1}{2} \left( x_{\text{lb},i}^* + x_{\text{ub},i}^* \right), \forall i \in \mathcal{N}. \qquad (9)$$

For integer features with a fractional mid-point between the upper and lower bound, we randomly select its floor or ceiling. For categorical features, we sample from the subset of available categories:

$$x_{\text{mid},i}^* = \text{uniform}(\mathbf{x}_{\text{cat}}^*), \forall i \in \mathcal{C}. \qquad (10)$$

For the unconstrained case, we use $\mathbf{x}_{\text{mid}}^*$ as the next query point $\mathbf{x}^*$. When additional input constraints $h(\mathbf{x})$ and/or $g(\mathbf{x})$ are given, we know that at least one point in $[\mathbf{x}_{\text{lb}}^*, \mathbf{x}_{\text{ub}}^*], \mathbf{x}_{\text{cat}}^*$ is feasible even if the heuristic $\mathbf{x}_{\text{mid}}^*$ is infeasible. To repair the solution $\mathbf{x}_{\text{mid}}^*$, we project it onto the feasible space:

$$\mathbf{x}^* \in \underset{\mathbf{x} \in [\mathbf{x}_{\text{lb}}^*, \mathbf{x}_{\text{ub}}^*], \mathbf{x}_{\text{cat}}^*}{\arg \min} \sum_{i \in \mathcal{N}} \left( x_{\text{mid},i}^* - x_i \right)^2 - \sum_{i \in \mathcal{C}} \sum_{j \in x_{\text{mid},i}^*} \nu_{i,j} \qquad (11a)$$

$$h(\mathbf{x}) = 0, \qquad\qquad\qquad (11b)$$

$$g(\mathbf{x}) \leq 0. \qquad\qquad\qquad (11c)$$

Eq. (11) projects $\mathbf{x}_{\text{mid}}^*$ onto the feasible set defined by $h(\mathbf{x})$ and $g(\mathbf{x})$. The time complexity of solving Eq. 11, and that of solving Eqs. (3)–(6) for the search space $[\mathbf{x}_{\text{lb}}^*, \mathbf{x}_{\text{ub}}^*], \mathbf{x}_{\text{cat}}^*$, are both NP-hard. However, in preliminary evaluations we found that Gurobi 9 often solves both problems to $\epsilon$-global optimality for moderately-sized tree models in less time compared to random sampling.

**Hyperparameters.** We train the kernel hyperparameters signal variance $\sigma_0^2$ and noise $\sigma_y^2$ by maximizing the log marginal likelihood. Additional hyperparameters are introduced by the gradient-boosted tree ensemble trained at every iteration, i.e., maximum tree depth and number of trees, and through $\kappa$ in the UCB. For the Section 5 numerical studies, we leave $\kappa$ and gradient-boosting hyperparameters constant and only increase maximum tree depth and number of trees for the high-dimensional CIFAR-NAS benchmark to capture more complicated interactions. The appendix reports specific values for all hyperparameters.

**Limitations.** The method suffers from standard BO challenges where the tree kernel may not be a good prior for the underlying black-box function, e.g., purely continuous feature spaces. More

limitations arise from solving an NP-hard problem to global optimality when working with large data sets in high-dimensional search spaces. For cases where solving the NP-hard problem is too difficult but BO is still applicable, Gurobi 9 can typically develop good feasible solutions as a heuristic.

## 5 Numerical studies

This section empirically evaluates the performance of tree kernel-based GPs using a wide variety of synthetic and real-world benchmark problems. We show (i) the ability of tree kernels to capture uncertainty of the underlying tree ensemble, (ii) the advantage of using global vs. local strategies for optimizing the acquisition function and (iii) the proposed algorithm's superior performance in cases with constrained search spaces and mixed variable types. `LEAF-GP` denotes the Section 4 proposed algorithm, and Section 2 outlines the baseline of methods we compare against. For every run, we visualize the median and confidence intervals of the first and third quartile based on 20 individual runs with varying random seeds. Further technical details can be found in the appendix.

### 5.1 Uncertainty metric

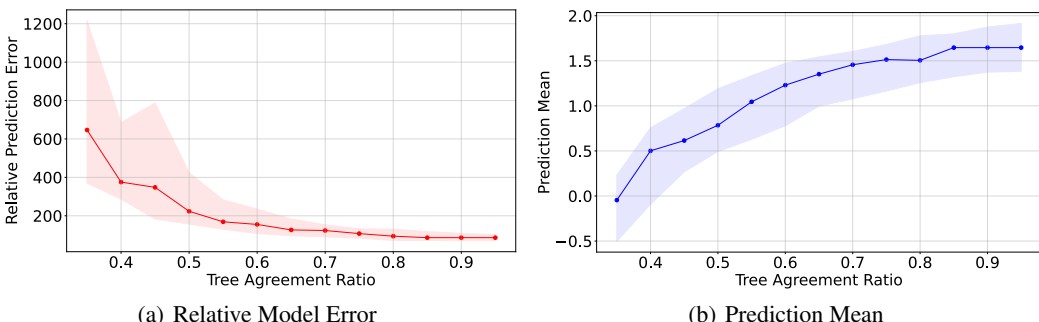

(a) Relative Model Error

(b) Prediction Mean

Figure 2: The relative prediction error (Eq. 13) and model prediction mean over the maximum tree agreement ratio $R$ for benchmark problem Rastrigin (10D). Changing $R$ is equivalent to changing the maximum kernel covariance. Plot shows the median line and confidence intervals (first and third quartile) from 20 random seeds. Section 5.1 provides more details.

The tree kernel-based GP uses the leaf overlap measure to quantify correlation between two inputs. In BO, such measures help identify unexplored areas where correlation to existing training data is low and we expect inaccurate model predictions. To empirically test the tree kernel's capability of identifying uncertainty in the underlying tree ensemble, we change the optimization formulation:

$$\mathbf{x}_{\text{lb}}^*, \mathbf{x}_{\text{ub}}^*, \mathbf{x}_{\text{cat}}^* \in \underset{\mathbf{x}, \mathbf{z}, \nu}{\arg\max} \; \mu(\mathbf{x}), \tag{12a}$$

$$\text{s.t. Eq. (3), Eq. (4), Eq. (6), Eq. (7), Eq. (8)} \tag{12b}$$

$$|\mathcal{T}|^{-1} \sum_{t \in \mathcal{T}} \sum_{l \in \mathcal{L}_t} A_{i,l} z_{t,l} \leq R, \; \forall i \in [m] \tag{12c}$$

Eq. (12a) maximizes over the mean prediction of the tree kernel-based GP with Eq. (12c) restricting the ratio of tree agreement for the black-box query point $\mathbf{x}^*$ defined as the Eq. (9) center of the optimal area $\mathbf{x}_{\text{lb}}^*, \mathbf{x}_{\text{ub}}^*, \mathbf{x}_{\text{cat}}^*$. A solution to problem Eq. (12) guarantees a maximum leaf overlap of $R$ between $\mathbf{x}^*$ and all available data points $\mathbf{X}$. Limiting the tree agreement ratio corresponds to constraining the Eq. (1) kernel correlation between the existing dataset and the proposed optimal area $\mathbf{x}_{\text{lb}}^*, \mathbf{x}_{\text{ub}}^*, \mathbf{x}_{\text{cat}}^*$. Here, we evaluate values of $R \in [0.35, 1.0]$ with increments of $0.05$ and compute the model error according to:

$$\epsilon_{\text{error}} = \left| \frac{\mu(\mathbf{x}^*) - f_{\text{true}}(\mathbf{x}^*)}{\mu(\mathbf{x}^*)} \right| \tag{13}$$

Fig. 2 shows results for the Rastrigin [63] benchmark. With smaller values of $R$, the Eq. (12) optimization problem becomes more restricted, leading to smaller objective values for solution $\mathbf{x}^*$. Fig. 2(a) shows that a growing leaf overlap (increasing $R$) reduces the model error defined in Eq. (13).

This suggests that the kernel works as intended and that low kernel correlation can reveal search space areas with high model uncertainty. The appendix provides results for additional benchmarks.

## 5.2 Local vs. global acquisition optimization

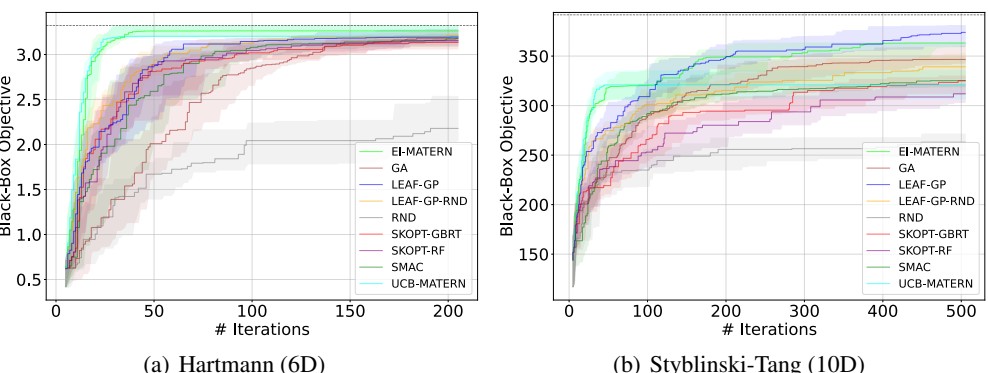

(a) Hartmann (6D)             (b) Styblinski-Tang (10D)

Figure 3: Black-box optimization progress of `LEAF-GP` vs. baseline. Plot shows the median line and confidence intervals (first and third quartile) from 20 random seeds. Section 5.2 provides more details

This section (i) compares `LEAF-GP` to other state-of-the-art algorithms for common benchmarks and (ii) shows the advantage of global vs. local strategies for optimizing the acquisition function. We introduce a variation of the proposed algorithm, `LEAF-GP-RND`, which optimizes the same acquisition function as `LEAF-GP`, but uses random sampling instead of Gurobi 9. Fig. 3 shows results for Hartmann (6D) [63] and Styblinski-Tang (10D) [63]. We do not expect tree model-based approaches to perform well since both benchmark functions are continuous. However, the benchmarks show that all approaches perform similarly for Hartmann (6D) with `UCB-MATERN` and `EI-MATERN` improving the black-box objective at the fastest rate. For Styblinski-Tang (10D), `LEAF-GP` significantly outperforms other algorithms. Although `LEAF-GP` is not specialized to this setting, observe that it performs roughly equivalently to the state of the art on these continuous, unconstrained optimization problems.

Although sampling-based `LEAF-GP-RND` and optimization-based `LEAF-GP` perform similarly on the smaller Hartmann (6D), `LEAF-GP` is particularly strong on the higher-dimensional Styblinski-Tang (10D) benchmark function, and two additional benchmarks in the appendix. Moreover, `LEAF-GP-RND` does not support explicit input constraints and performs particularly bad for BO with known constraints. For the following constrained benchmarks, we remove `LEAF-GP-RND` from the comparison. The appendix has additional details and results.

## 5.3 Constrained spaces

This section presents numerical benchmarks with known input constraints. The acquisition optimization strategy of `LEAF-GP` allows explicit consideration of input constraints, i.e., logical and convex/non-convex $n$-th degree polynomial equality and inequality constraints for mixed variable spaces. The `UCB-MATERN` and `EI-MATERN` implementations of BoTorch supports only linear equality and inequality constraints at the acquisition optimization step. The `GA` pymoo algorithm has an interface for callable constraint functions which are considered when generating new generations of candidate points. When a method does not support the specific input constraints, we penalize the objective:

$$f_{\text{penalty}} = \lambda \left( \max(g(\mathbf{x}), 0)^2 + h(\mathbf{x})^2 \right), \tag{14}$$

where $g(\mathbf{x}) \leq 0$ and $h(\mathbf{x}) = 0$ are inequality and equality constraints, respectively. This penalty strategy allows methods that do not support explicit constraints to still produce feasible points given enough iterations. Eq. (14) introduces the hyperparameter $\lambda$ which weights the penalty, we test values $\lambda \in \{1, 10, 100\}$ and only plot the best run for all methods that rely on constraint penalization. To initialize every method with feasible points, we draw random samples from a uniform distribution within the bounds that define the search space and compute the closest feasible point similar to Eq. (11). For constrained benchmarks, we introduce `FEAS-RANDOM` which simply projects random

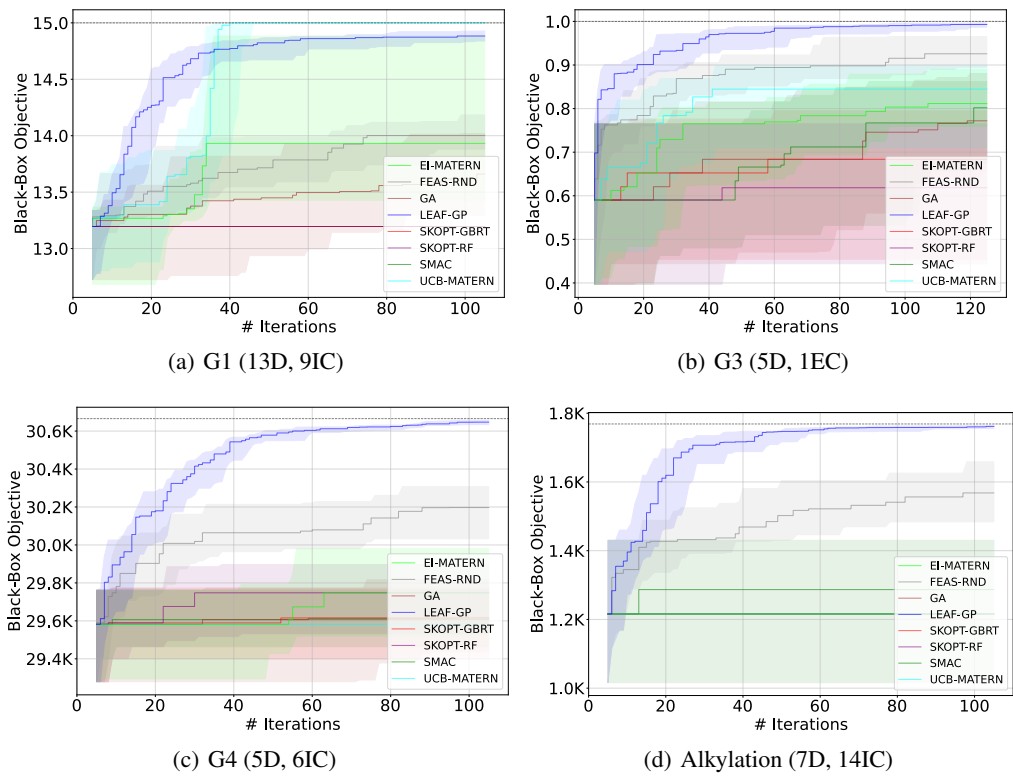

(a) G1 (13D, 9IC)  (b) G3 (5D, 1EC)

(c) G4 (5D, 6IC)  (d) Alkylation (7D, 14IC)

Figure 4: Feasible black-box optimization progress of `LEAF-GP` vs. baseline. Plot shows the median line and confidence intervals (first and third quartile) from 20 random seeds. Confidence intervals are neglected for methods that cannot improve the initial training data. Figure subtitles give the function name and number of: dimensions (D), equality constraints (EC), and inequality constraints (IC). Section 5.3 provides more details.

samples onto the set of feasible points that satisfy $h(\mathbf{x})$ and $g(\mathbf{x})$. Fig. 4 plots feasible solutions to four different continuous benchmark problems, G1, G3, G4 [32], and Alkylation [60]. The G1 benchmark has linear inequality constraints only, which are supported by `UCB-MATERN`, `EI-MATERN` and `LEAF-GP`. Fig. 4(a) shows `LEAF-GP` making quick progress at the beginning with `UCB-MATERN` catching up towards the end. G3 and G4 have different combinations of nonlinear constraints, and `LEAF-GP` significantly outperforms competing methods and random feasible sampling. The Alkylation benchmark determines optimal operating conditions for a simplified alkylation process considering complicated nonlinear constraints representing economic, physical and performance limits. Again, `LEAF-GP` outperforms other methods, which often struggle to find feasible solutions. More details regarding the presented benchmark problems are given in the appendix.

## 5.4 Mixed-variable spaces

We now consider spaces that exhibit mixtures of continuous, integer and categorical variables for constrained problems. Tree model-based algorithms naturally support categorical variables by replacing continuous splits with categorical splits. For all methods that do not support categorical features, we use one-hot encoding. The category with the highest corresponding auxiliary variable value is chosen for the subsequent query point. On the Pressure Vessel (4D) [12] benchmark, which comprises two continuous features, two integer features and three inequality constraints, Fig. 5(a) shows that `LEAF-GP` outperforms other black-box optimization methods.

The CIFAR-NAS (29D) is a high-dimensional benchmark problem with one continuous, 23 integer and five categorical variables. It describes properties of different layers and training hyperparameters for a convolutional neural network (CNN) trained on CIFAR-10 [40]. The problem's search space is hierarchical, as different layers, i.e., convolutional and fully-connected layers, can be activated.

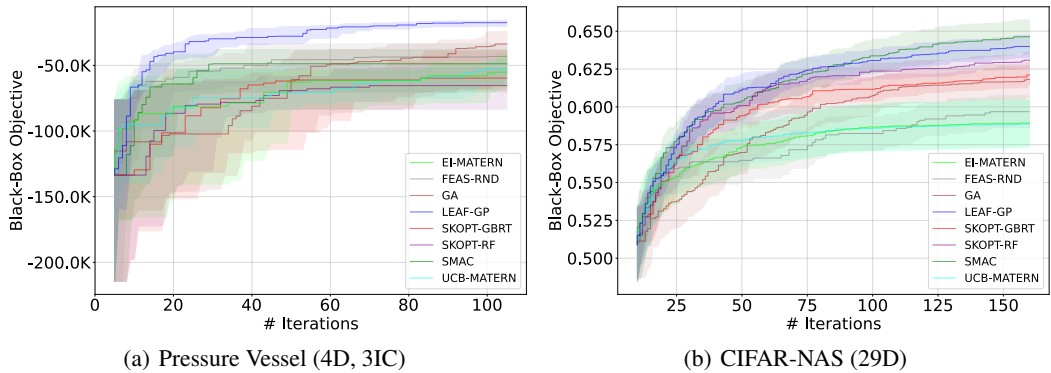

(a) Pressure Vessel (4D, 3IC)     (b) CIFAR-NAS (29D)

Figure 5: Feasible black-box optimization comparing `LEAF-GP` vs. baseline. Plot shows median line and confidence intervals (first and third quartile) from 20 random seeds. Figure subtitles give the number of dimensions (D) and inequality constraints (IC). Section 5.4 provides details.

Properties describing a layer are only relevant if the layer is active. For this problem, we introduce two types of constraints to guide the `LEAF-GP`: (i) constraints that allow for feasible network designs, i.e., different values for padding, stride, kernel size and max pooling in earlier layers affects what kernel size is feasible in following layers and (ii) constraints that ensure that hierarchies are respected. Towards the former, we include constraints to ensure the output size of every convolutional layer is positive. In general, tree models are particularly good at capturing hierarchical search space structures. To enforce hierarchical relationships during the acquisition optimization step, we introduce indicator constraints that force layer properties to take predefined default values if the layer is inactive. This ensures that the optimizer avoids leaves that infer splitting conditions from feature properties of inactive layers, thus effectively reducing the search space. To expedite tests, we train the CNN on half of the CIFAR-10 training data and optimize for test accuracy. Fig. 5(b) summarizes the results of this study and shows that tree model-based algorithms generally outperform `UCB-MATERN` and `EI-MATERN`. Utilizing search space constraints, `SMAC` and `LEAF-GP` find the neural architectures with the highest test accuracy with `SMAC` slightly outperforming `LEAF-GP`. Since finding feasible architectures for CIFAR-10 is not particularly challenging, Fig. 9 in the appendix shows a benchmark for tuning variational autoencoders (VAE) adapted from Daxberger et al. [16], where `LEAF-GP` significantly outperforms other algorithms. Finding feasible VAE architectures is more difficult given the requirement that the latent encoding must be decoded back to original input image size.

# 6 Conclusion

We present a framework for black-box optimization based on tree kernel Gaussian processes that simultaneously allows (i) reliable uncertainty quantification of mixed feature spaces and (ii) incorporation of explicit input constraints. Although these two needs have been considered separately, we are able to address both simultaneously through the mixed-integer second-order cone formulation of the acquisition function. The numerical studies show that the proposed strategy performs competitively with state-of-the-art algorithms for unconstrained problems and may significantly outperform existing methods for constrained benchmarks, especially those with mixed feature spaces. We use optimization constraints together with the acquisition functions to incorporate domain knowledge and leverage hierarchical search space structures, e.g., for neural architecture search.

# 7 Acknowledgements

This work was supported by BASF SE, Ludwigshafen am Rhein, EPSRC Research Fellowships to R.M. (EP/P016871/1) and CT (EP/T001577/1), and an Imperial College Research Fellowship to CT.

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
