# A  Appendix

# B  General experimental setup

All experimental results presented in Section 5 were evaluated on an HTCondor cluster (see [64]) of machines equipped with Intel Core i7-8700 3.20GHz and 16 GB RAM. Confidence intervals show the first and third quartile of 20 independent runs with random seeds $\in [101, 102, \ldots, 120]$. If not specifically indicated in Section C, all competing algorithms use default values for all hyperparameters. To allow for a fair comparison, we give the same set of initial data points to all tested methods for both constrained and unconstrained benchmark problems. A set of ten initial samples is used for the Section 5.4 CIFAR-NAS example due to its high dimensionality. Five initial points are used for the remaining benchmark problems.

# C  Algorithms

This section summarizes the different algorithms used for the Section 5 numerical studies. We give implementation details and hyperparameter settings to reproduce the presented results.

## C.1  LEAF-GP and LEAF-GP-RND

The `LEAF-GP` uses LightGBM [37] for training gradient-boosted tree ensembles. All runs use the hyperparameter value *min_data_in_leaf* $= 1$ as the training dataset size needs to be at least twice the minimum number of data points a leaf is based on. The *min_data_in_leaf* default value of LightGBM is 20, which would cause run-time errors after initialization. We also set LightGBM hyperparameter *min_data_per_group* $= 1$. For the high-dimensional benchmark problem CIFAR-NAS, we set *max_depth* $= 5$ and *num_boost_rounds* $= 100$ for training the ensemble in LightGBM, referring to maximum interaction depth per decision tree and total number of trees in the ensemble, respectively. For all other benchmarks we use *max_depth* $= 3$ and *num_boost_rounds* $= 50$.

We implement the tree ensemble kernel as a non-stationary kernel in GPyTorch [24]. For deriving the posterior distribution, we use a Gaussian likelihood and standardize the target values of the data set. Section 3 introduces signal variance $\sigma_0$ and noise term $\sigma_y$ as kernel hyperparameters which are fitted by maximizing the marginal log likelihood over 200 epochs using the Adam solver [38]. The hyperparameters are constrained by intervals according to $\sigma_0 \in [5e{-}4, 0.2]$ and $\sigma_y \in [0.05, 20.0]$.

For `LEAF-GP` the Section 4 acquisition optimization formulation is encoded using *gurobipy* [28] and solved using Gurobi 9. Runs are limited to 100 s if the solver finds a feasible solution and are continued otherwise.

Moreover, we set *heuristics* $= 0.2$ and activate the non-convex hyperparameter for benchmark problems with non-convex constraints. `LEAF-GP-RND` uses a sampling-based strategy that randomly evaluates the acquisition function at 2000 locations and selects the maximimum value.

## C.2  GA

We use the standard Genetic Algorithm implementation of the pymoo [7] toolbox for evolutionary algorithms and change *population_size* $= 10$ given the small evaluation budget. Default values are used for all other hyperparameters.

## C.3  SKOPT-GBRT and SKOPT-RF

We use the default implementation of Scikit-Optimize [31] with random forest and gradient-boosted trees base estimators for `SKOPT-RF` and `SKOPT-GBRT`, respectively. Default values are used for all hyperparameters.

## C.4  SMAC

For `SMAC` [34] we utilize the most recent Python implementation SMAC3 [44] using random forest models. Moreover, we specify the hyperparameters *run_obj* $=$ '*quality*' and activate the deterministic

flag to allow for reproducibility. The Section 5.4 CIFAR-NAS and Section F VAE-NAS benchmark problems exhibit hierarchical search space relationships, i.e., hyperparameters of a specific layer are only relevant if the layer is active. We use `SMAC`'s `InCondition` function which allows for certain child features to be considered only if some parent features have certain values, e.g., the stride of layer $n$ is only considered if number of layers is at least $n$. This allows `SMAC` to capture hierarchical relationships explicitly and to avoid evaluating multiple equivalent configurations. Default values are used for all other hyperparameters.

### C.5   UCB-MATERN and EI-MATERN

The `UCB-MATERN` and `EI-MATERN` algorithms use the standard upper confidence bound and expected improvement implementations of BoTorch [2]. Before fitting a GP, we normalize data features and standardize data outputs. The upper confidence acquisition hyperparameter $\beta$ is set to 1.96. We negate the target values and define *raw_samples* $= 200$ for the acquisition function maximization. For unconstrained cases, the acquisition optimizer uses 100 restarts. However, for constrained problems we limit the optimizer to five restarts due to extensive run-times caused by finding feasible solutions. Default values are used for all other hyperparameters.

## D   Benchmark problems

### D.1   Unconstrained problems

Figure 3 shows the results for Hartmann (6D), Rastrigin (10D), Schwefel (10D) and Styblinski-Tang (10D) benchmark functions implemented according to Surjanovic and Bingham [63]. Table 1 summarizes number of dimensions and domain evaluated for unconstrained benchmark problems.

Table 1: Benchmark functions for local vs. global acquisition-function optimization tests. Table shows function name, number of dimensions (Dim.), and domain of input variables

| Function | Dim. | Domain |
|---|---|---|
| Hartmann | 6 | $\mathbf{x} \in [0.0, 1.0]^6$ |
| Rastrigin | 10 | $\mathbf{x} \in [-4.0, 5.0]^{10}$ |
| Schwefel | 10 | $\mathbf{x} \in [-500.0, 500.0]^{10}$ |
| Styblinski-Tang | 10 | $\mathbf{x} \in [-5.0, 5.0]^{10}$ |

### D.2   Constrained problems

Fig. 8 presents results of benchmark problems with known constraints. Domain bounds without decimals indicate integer-valued variable types. Benchmark examples G1, G3, G4, G6, G7 and G10 are implemented according to Hedar [32]. The Alkylation benchmark [60] is adapted from an open-source implementation [62]. To compare methods that do not support specific input constraints, we penalize the black-box function output according to:

$$f_{\text{penalty}} = \lambda \left( \max(g(\mathbf{x}), 0)^2 + h(\mathbf{x})^2 \right), \tag{15}$$

where $g(\mathbf{x}) \leq 0$ and $h(\mathbf{x}) = 0$ are inequality and equality constraints, respectively. Maximizing the combined black-box output allows methods to find feasible solutions. Eq. (15) has the hyperparameter $\lambda$ which weights the penalty, we test values $\lambda \in \{1, 10, 100\}$ and only plot the best run for all methods that rely on constraint penalization. In the tests conducted, `LEAF-GP` fully supports explicit input constraints. `UCB-MATERN` and `EI-MATERN` support linear inequality and equality constraints only. The evolutionary algorithm `GA` has built-in constraint consideration but does not guarantee feasible solutions. `LEAF-GP-RND`, `SKOPT-GBRT`, `SKOPT-RF` and `SMAC` rely on the Eq. (15) penalty function.

### D.3   CIFAR-NAS

Table 3 gives more details on the Section 5.4 CIFAR-NAS (29D) benchmark problem. CIFAR-NAS (29D) has a total of 29 hyperparameters to tune, i.e., 1 continuous, 15 integer, 8 binary and

Table 2: Benchmark functions for constrained search space tests. Table shows function name and number of: dimensions (D), equality constraints (EC), and inequality constraints (IC). Values in brackets indicate the number of linear constraints which are natively supported by some of the algorithms. Domain bounds without decimals indicate integer-valued variables.

| Function | D | IC | EC | Domain |
|---|---|---|---|---|
| G1 | 13 | 9 (9) | 0 (0) | $\mathbf{x}_{\{0,...,8,12\}} \in [0.0, 1.0]$, $\mathbf{x}_{\{9,10,11\}} \in [0.0, 100.0]$ |
| G3 | 5 | 0 (0) | 1 (0) | $\mathbf{x} \in [0.0, 1.0]^5$ |
| G4 | 5 | 6 (0) | 0 (0) | $\mathbf{x}_0 \in [78.0, 102.0]$, $\mathbf{x}_1 \in [33.0, 45.0]$, $\mathbf{x}_{\{2,3,4\}} \in [27.0, 45.0]$ |
| G6 | 2 | 2 (0) | 0 (0) | $\mathbf{x}_0 \in [13.0, 100.0]$, $\mathbf{x}_1 \in [0.0, 100.0]$ |
| G7 | 10 | 8 (3) | 0 (0) | $\mathbf{x} \in [-10.0, 10.0]^{10}$ |
| G10 | 8 | 6 (3) | 0 (0) | $\mathbf{x}_0 \in [100.0, 10.0\mathrm{K}]$, $\mathbf{x}_{\{1,2\}} \in [1.0\mathrm{K}, 10.0\mathrm{K}]$, $\mathbf{x}_{\{3,...7\}} \in [10.0, 1.0\mathrm{K}]$ |
| Alkylation | 7 | 14 (0) | 0 (0) | $\mathbf{x}_0 \in [0.0, 2.0\mathrm{K}]$, $\mathbf{x}_1 \in [0.0, 16.0\mathrm{K}]$, $\mathbf{x}_2 \in [0.0, 120.0]$, $\mathbf{x}_3 \in [0.0, 5.0\mathrm{K}]$, $\mathbf{x}_4 \in [90.0, 95.0]$, $\mathbf{x}_5 \in [0.01, 4.0]$, $\mathbf{x}_6 \in [145.0, 162.0]$ |
| Pressure Vessel | 4 | 3 (2) | 0 (0) | $\mathbf{x}_{\{0,1\}} \in [1, 99]$, $\mathbf{x}_{\{2,3\}} \in [10.0, 200.]$ |

5 categorical variables. The goal is to select hyperparameter values for a CNN trained in PyTorch [55] on the CIFAR-10 dataset [40] that maximize test accuracy. The training and test scripts were adapted from Trencseni [70]. Due to limited computing resources, we train the CNN on half of the training data for 10 epochs using the Adam solver [38]. We score networks using the full test data set. Only certain combinations of stride, padding and filter size for various layers result in feasible neural architectures, e.g., the filter size of one layer may be too large given the output of the previous layer. In such cases the CNN training fails, and the black-box returns the largest black-box value found so far, helping algorithms learn to avoid infeasible neural architectures. To simplify the training, layer inputs are parameterized based on outputs of the previous layer for convolutional and fully-connected layers, as well as the intermediate connecting layer. The benchmark introduces categorical variables for activation function selection. Methods that do not support categorical features use one-hot encoding.

`LEAF-GP` has access to constraints capturing feasible neural architectures mainly concerned with the convolutional layers. Algorithms can choose to activate at most three convolutional and two fully-connected layers. To capture constraints for feasible CNNs, we introduce $w_{\mathrm{out},i}$ as the output of convolutional layer $i$ and $W_{\mathrm{out},i}$ as the layer's modified output in case max-pooling is applied:

$$w_{\mathrm{out},i} \in \mathbb{N}_0, \forall i \in \{1, 2, 3\} \tag{16a}$$

$$W_{\mathrm{out},i} \in \mathbb{N}_0, \forall i \in \{1, 2, 3\} \tag{16b}$$

Convolutional layers use PyTorch's `Conv2D` with inputs derived by the optimization algorithms. PyTorch's `MaxPool2d` implements the max-pooling with the commonly-used $(2, 2)$ kernel size.

$$W_{\mathrm{in},1} = 32 \tag{17a}$$

$$w_{\mathrm{out},1} = \frac{W_{\mathrm{in},1} - F_1 + 2P_1}{S_1} + 1 \tag{17b}$$

$$W_{\mathrm{out},1} = b_1^{\mathrm{conv}} \lfloor w_{\mathrm{out},1}(1 - 0.5b_1^{\mathrm{pool}}) \rfloor + (1 - b_1^{\mathrm{conv}})W_{\mathrm{in},1} \tag{17c}$$

$$w_{\mathrm{out},2} = \frac{W_{\mathrm{out},1} - F_2 + 2P_2}{S_2} + 1 \tag{17d}$$

$$W_{\mathrm{out},2} = b_2^{\mathrm{conv}} \lfloor w_{\mathrm{out},2}(1 - 0.5b_2^{\mathrm{pool}}) \rfloor + (1 - b_1^{\mathrm{conv}})W_{\mathrm{out},1} \tag{17e}$$

$$w_{\text{out},3} = \frac{W_{\text{out},2} - F_3 + 2P_3}{S_3} + 1 \tag{17f}$$

$$W_{\text{out},3} = b_3^{\text{conv}} \lfloor w_{\text{out},3}(1 - 0.5b_3^{\text{pool}}) \rfloor + (1 - b_3^{\text{conv}})W_{\text{out},2} \tag{17g}$$

$$W_{\text{out},3} \geq 1 \tag{17h}$$

$$1 \leq b_1^{\text{conv}} + b_2^{\text{conv}} + b_3^{\text{conv}} + b_1^{\text{fc}} + b_2^{\text{fc}} \tag{17i}$$

The $32 \times 32$ image size of CIFAR-10 data defines the input to the full CNN ($W_{\text{in},1}$) in Eq. (17a). Eq. (17b) combines filter size $F_1$, padding $P_1$ and stride $S_1$ of the first convolutional layer to compute its output size $w_{\text{out},1}$. Variable $W_{\text{out},1}$ captures the final output of the convolutional layer by considering if the layer is activated, i.e., $b_1^{\text{conv}} = 1$, and if max-pooling is applied, i.e., $b_1^{\text{pool}} = 1$. Constraints (17c)–(17g) denote the same restrictions for subsequent layers, each using the output size of the previous layer as its input size. Eq. (17h) ensures that the output of the last convolutional layer $W_{\text{out},3}$ is at least one. We also enforce that at least one layer be active, which is captured by Eq. (17i).

To break symmetries in the benchmark problems, we introduce Constraints (18a)–(18g):

$$b_3^{\text{conv}} \leq b_2^{\text{conv}} \leq b_1^{\text{conv}} \tag{18a}$$

$$\neg b_i^{\text{conv}} \rightarrow \neg b_i^{\text{pool}}, \forall i \in \{1,2,3\} \tag{18b}$$

$$\neg b_i^{\text{conv}} \rightarrow C_i^{\text{conv}} \leq 4, \forall i \in \{1,2,3\} \tag{18c}$$

$$\neg b_i^{\text{conv}} \rightarrow F_i \leq 2, \forall i \in \{1,2,3\} \tag{18d}$$

$$\neg b_i^{\text{conv}} \rightarrow S_i \leq 1, \forall i \in \{1,2,3\} \tag{18e}$$

$$\neg b_i^{\text{conv}} \rightarrow P_i \leq 0, \forall i \in \{1,2,3\} \tag{18f}$$

$$\neg b_i^{\text{conv}} \rightarrow Act_i^{\text{conv}} = \text{ReLU}, \forall i \in \{1,2,3\} \tag{18g}$$

Eq. (18a) activates layers in a particular order, and Eq. (18b) deactivates max-pooling when the associated convolutional layer is inactive. Constraints (18c)–(18g) set layer-specific hyperparameters to pre-defined default values when the associated layer is inactive. We select these defaults as the lower bound for non-categorical variables and the first category for categorical variables.

Constraints (19) express the same restrictions for fully-connected layers:

$$b_2^{\text{fc}} \leq b_1^{\text{fc}} \tag{19a}$$

$$\neg b_i^{\text{fc}} \rightarrow N_i^{\text{fc}} \leq 4, \forall i \in \{1,2\} \tag{19b}$$

$$\neg b_i^{\text{fc}} \rightarrow Act_i^{\text{fc}} = \text{ReLU}, \forall i \in \{1,2\} \tag{19c}$$

Table 3: Table shows all hyperparameter names, types and domains of the CIFAR-NAS benchmark. The transformation column refers to post-processing computations before passing the hyperparameter value to the neural network training.

| # | Name | Type | Domain | Transformation |
|---|------|------|--------|----------------|
| 0 | Batch size | integer | $[2, 4]$ | $N_{\text{batch}} = 2^{x_0}$ |
| 1 | Learning rate | conti. | $[-5.0, -1.0]$ | $\alpha = 10^{x_1}$ |
| | **Convolutional layer 1** | | | |
| 2 | Layer is active | binary | $\{0, 1\}$ | $b_1^{\text{conv}} = x_2$ |
| 3 | Number of channels | integer | $[2, 4]$ | $C_1^{\text{conv}} = 2^{x_3}$ |
| 4 | Max pooling is active | binary | $\{0, 1\}$ | $b_1^{\text{pool}} = x_4$ |
| 5 | Filter size | integer | $[2, 5]$ | $F_1 = x_5$ |
| 6 | Stride | integer | $[1, 3]$ | $S_1 = x_6$ |
| 7 | Padding | integer | $[0, 3]$ | $P_1 = x_7$ |
| 8 | Activation function | categ. | $\{\text{ReLU, PReLU, Leaky ReLU}\}$ | $Act_1^{\text{conv}} = x_8$ |
| | **Convolutional layer 2** | | | |
| 9 | Layer is active | binary | $\{0, 1\}$ | $b_2^{\text{conv}} = x_9$ |
| 10 | Number of channels | integer | $[2, 4]$ | $C_2^{\text{conv}} = 2^{x_{10}}$ |
| 11 | Max pooling is active | binary | $\{0, 1\}$ | $b_2^{\text{pool}} = x_{11}$ |
| 12 | Filter size | integer | $[2, 5]$ | $F_2 = x_{12}$ |
| 13 | Stride | integer | $[1, 3]$ | $S_2 = x_{13}$ |
| 14 | Padding | integer | $[0, 3]$ | $P_2 = x_{14}$ |
| 15 | Activation function | categ. | $\{\text{ReLU, PReLU, Leaky ReLU}\}$ | $Act_2^{\text{conv}} = x_{15}$ |
| | **Convolutional layer 3** | | | |
| 16 | Layer is active | binary | $\{0, 1\}$ | $b_3^{\text{conv}} = x_{16}$ |
| 17 | Number of channels | integer | $[2, 4]$ | $C_3^{\text{conv}} = 2^{x_{17}}$ |
| 18 | Max pooling is active | binary | $\{0, 1\}$ | $b_3^{\text{pool}} = x_{18}$ |
| 19 | Filter size | integer | $[2, 5]$ | $F_3 = x_{19}$ |
| 20 | Stride | integer | $[1, 3]$ | $S_3 = x_{20}$ |
| 21 | Padding | integer | $[0, 3]$ | $P_3 = x_{21}$ |
| 22 | Activation function | categ. | $\{\text{ReLU, PReLU, Leaky ReLU}\}$ | $Act_3^{\text{conv}} = x_{22}$ |
| | **Fully-connected layer 1** | | | |
| 23 | Layer is active | binary | $\{0, 1\}$ | $b_1^{\text{fc}} = x_{23}$ |
| 24 | Number of nodes | integer | $[2, 7]$ | $N_1^{\text{fc}} = 2^{x_{24}}$ |
| 25 | Activation function | categ. | $\{\text{ReLU, PReLU, Leaky ReLU}\}$ | $Act_1^{\text{fc}} = x_{25}$ |
| | **Fully-connected layer 2** | | | |
| 26 | Layer is active | binary | $\{0, 1\}$ | $b_2^{\text{fc}} = x_{26}$ |
| 27 | Number of nodes | integer | $[2, 7]$ | $N_2^{\text{fc}} = 2^{x_{27}}$ |
| 28 | Activation function | categ. | $\{\text{ReLU, PReLU, Leaky ReLU}\}$ | $Act_2^{\text{fc}} = x_{28}$ |

# E   Additional Results

This section presents additional results supporting the numerical evaluation in Section 5.

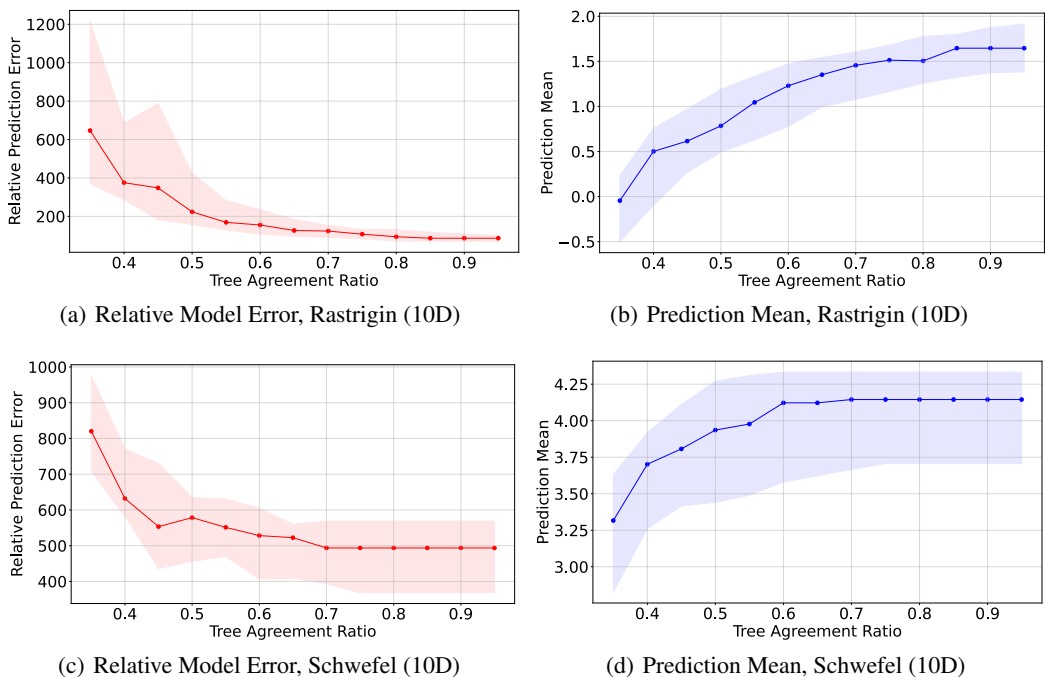

(a) Relative Model Error, Rastrigin (10D)

(b) Prediction Mean, Rastrigin (10D)

(c) Relative Model Error, Schwefel (10D)

(d) Prediction Mean, Schwefel (10D)

Figure 6: The relative prediction error (Eq. 13) and model prediction mean over the maximum tree agreement ratio $R$ for benchmark problem Schwefel (10D). Changing $R$ is equivalent to changing the maximum kernel covariance. Plot shows the median line and confidence intervals (first and third quartile) from 20 random seeds. Section 5.1 provides more details.

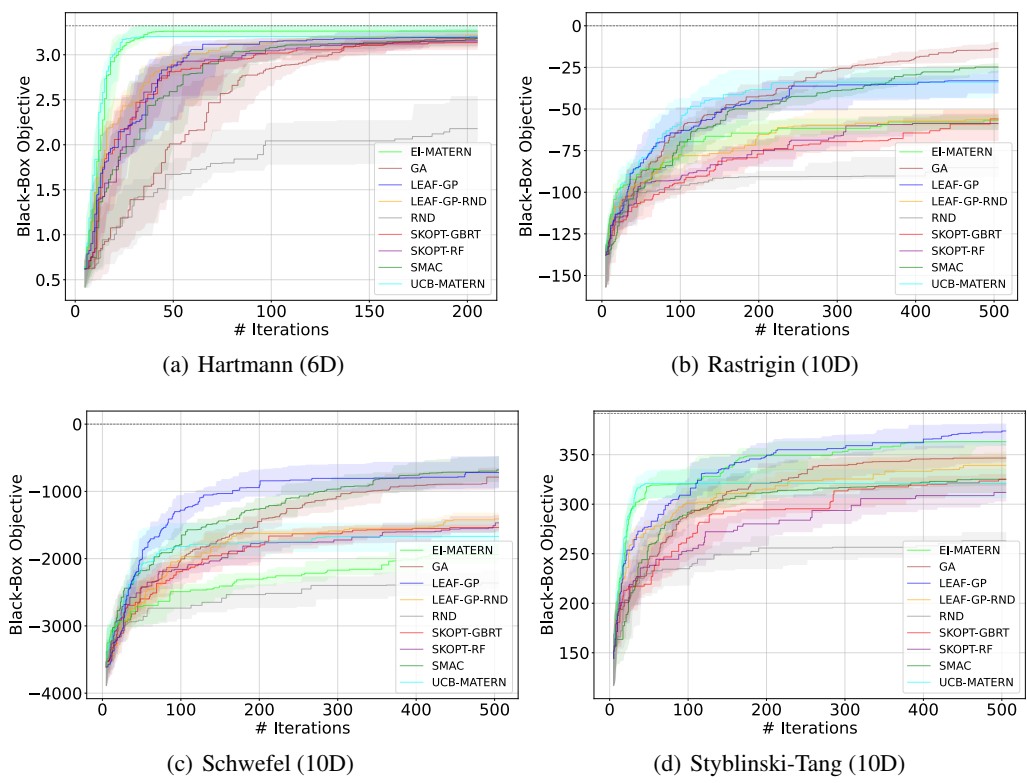

Figure 7: Black-box optimization progress of `LEAF-GP` vs. baseline. Plot shows the median line and confidence intervals (first and third quartile) from 20 random seeds. Section 5.2 provides more details

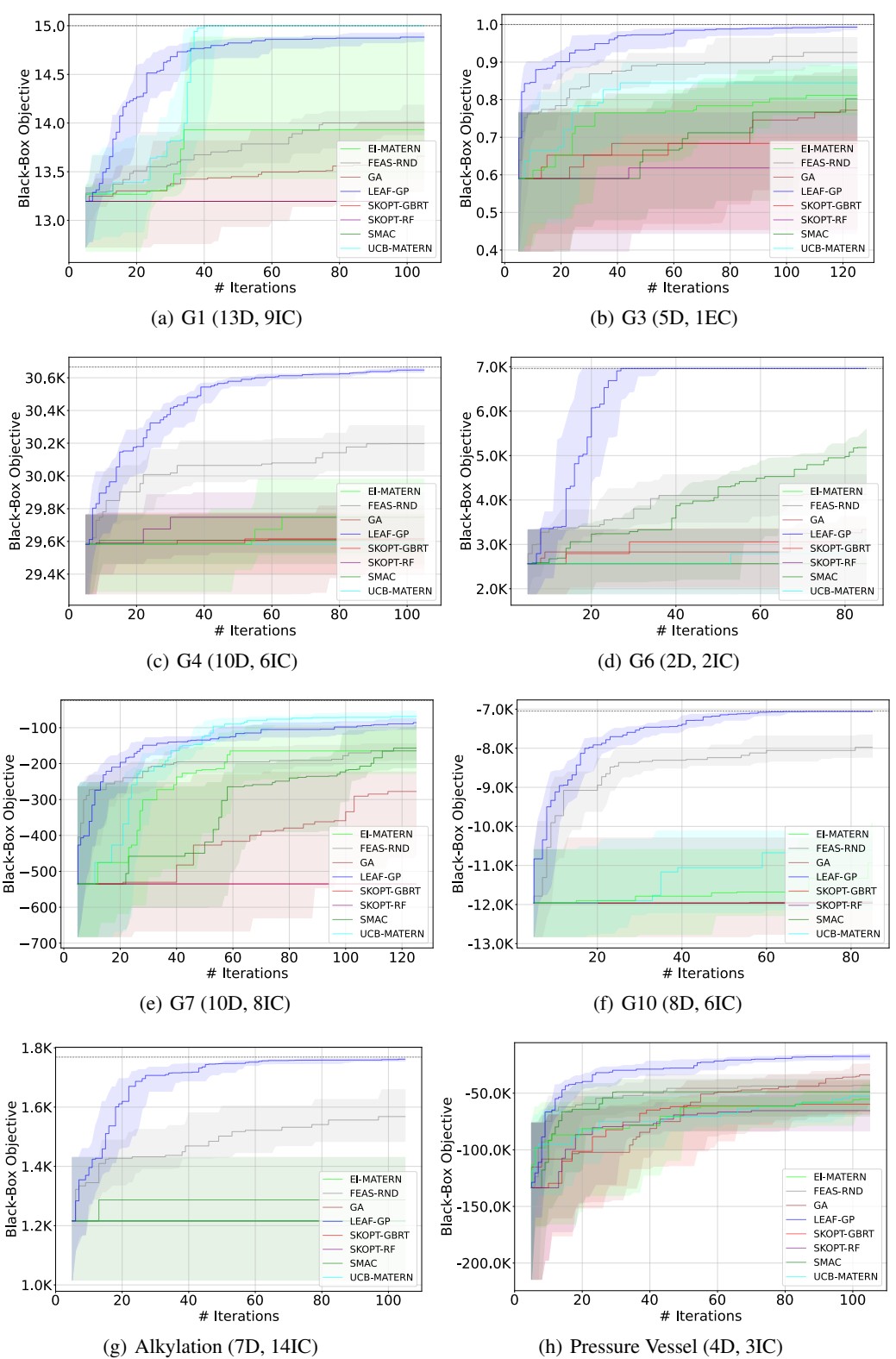

Figure 8: Feasible black-box optimization progress of `LEAF-GP` vs. baseline. Plot shows the median line and confidence intervals (first and third quartile) from 20 random seeds. Confidence intervals are neglected for methods that cannot improve the initial training data. Figure subtitles give the function name and number of: dimensions (D), equality constraints (EC), and inequality constraints (IC). Section 5.3 provides more details.

## F  VAE-NAS

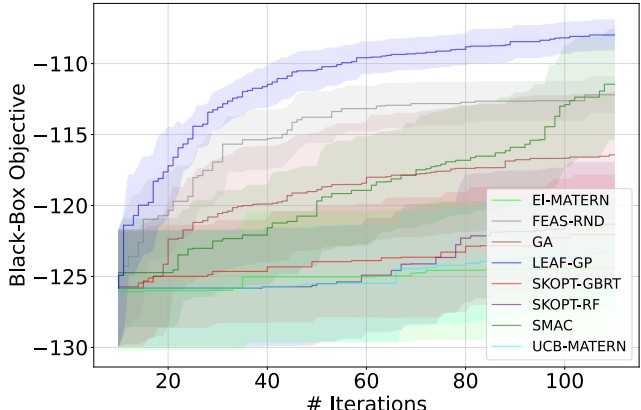

Figure 9: Feasible black-box optimization comparing `LEAF-GP` vs. baseline. Plot shows median line and confidence intervals (first and third quartiles) from 20 random seeds. Figure subtitles give the number of dimensions (D) and inequality constraints (IC). Section 5.4 provides details.

Table 4 gives more details on the Section 5.4 Variational Autoencoder Neural Architecture Search (VAE-NAS) benchmark problem, which was adapted from Daxberger et al. [16]. As their exact implementation is not publicly available, we created a benchmark problem based on the paper description and training scripts by Rath [58]. VAE-NAS (32D) has a total of 32 hyperparameters to tune, i.e., 1 continuous, 20 integer, and 11 categorical variables. The goal is to select hyperparameter values for a variational autoencoder (VAE) [39] trained in PyTorch [55] on the MNIST dataset [17] that minimize test loss, i.e., the average loss of when encoding and decoding images from the test set. We train each tested VAE for 32 epochs using the Adam solver [38] and a fixed batch size of 128. Only certain combinations of stride, padding, and filter size for various layers result in feasible neural architectures. The convolutional layer output size $W_{\text{out}}^e$ is computed as:

$$W_{\text{out}}^e = \frac{W_{\text{in}}^e - F^{\text{e}} + 2P^{\text{e}}}{S^{\text{e}}} + 1 \in \mathbb{N}, \tag{20}$$

Decoder constraints are more complicated given that the size of the VAE output must match the original MNIST image size. The output of deconvolutional layers is computed according to:

$$W_{\text{out}}^d = S^{\text{d}}(W_{\text{in}}^d - 1) + F^{\text{d}} - 2P^{\text{d}} + O^{\text{d}} \tag{21}$$

Matching the output and input sizes of the VAE is non-trivial, as hyperparameters of the convolutional and deconvolutional layers are themselves set by the optimization algorithms. To simplify the training, layer inputs are parameterized based on outputs of the previous layer for convolutional, deconvolutional, and fully-connected layers. Specifically, fully-connected layers `FC2` and `FC3` have $2 \times N^{\text{lat}}$ nodes. According to Daxberger et al. [16], we parameterize the size of the last fully-connected layer—which can be `FC3`, `FC4`, or the latent space layer depending on which layers are active—as $C_1^{\text{d}} \times 7 \times 7$. This allows methods not supporting explicit input constraints to still easily find a feasible neural architecture by deactivating all deconvolutional layers and computing an output $16 \times 7 \times 7 = 784$ with $C_1^{\text{d}} = 16$ (the original MNIST image size is $1 \times 28 \times 28 = 784$). This rule supersedes other layer size definitions for fully-connected layers. Moreover, we extend the benchmark by introducing the nonlinear activation function of each layer as a categorical optimization hyperparameter. Methods that do not support categorical features use one-hot encoding. The activation function of the last layer is fixed as the sigmoid function, superseding other activation function hyperparameters. The VAE is trained using the sum of binary cross-entropy loss (reconstruction error) and KL divergence. We use the same loss function to evaluate the VAE's performance on the test dataset, giving the black-box objective in Fig. 9. To allow for a fair comparison, the same 10 randomly sampled feasible architectures initialize all methods.

`LEAF-GP` has access to constraints describing feasible neural architectures. We begin by defining auxiliary variables similar to the benchmark in Section D.3:

$$W_{\text{in},i}^{\text{e}} \in \mathbb{N}_0, \qquad\qquad\qquad \forall i \in [1, 2], \tag{22a}$$

$$w^{\text{e}}_{\text{out},i} \in \mathbb{N}_0, \qquad\qquad \forall i \in [1,2], \quad (22\text{b})$$

$$W^{\text{e}}_{\text{out},i} \in \mathbb{N}_0, \qquad\qquad \forall i \in [1,2], \quad (22\text{c})$$

$$b^{\text{e}}_{\text{covn},i} \in \{0,1\}, \qquad\qquad \forall i \in [1,2], \quad (22\text{d})$$

$$N^{\text{e}}_{\text{conv}} = b^{\text{e}}_{\text{conv},1} + b^{\text{e}}_{\text{conv},2}, \qquad\qquad (22\text{e})$$

$$b^{\text{e}}_{\text{conv},1} \geq b^{\text{e}}_{\text{conv},2} \qquad\qquad (22\text{f})$$

where $W^{\text{e}}_{\text{in},i}$ and $W^{\text{e}}_{\text{out},i}$ denote, respectively, the input and output sizes of convolutional layer $i$ in the encoder. We also define an auxiliary variable $w^{\text{e}}_{\text{out},i}$ to track the output size for inactive layers, as well as binary variables $b^{\text{e}}_{\text{conv},i}$ corresponding to the active/inactive state of each layer. Eq. (22e) and Eq. (22f) link binary variables $b^{\text{e}}_{\text{conv},i}$ to the number of active convolutional layers in the encoder. Using these auxiliary variables, the following relations can be expressed:

$$W^{\text{e}}_{\text{in},1} = 28, \qquad\qquad (23\text{a})$$

$$W^{\text{e}}_{\text{in},2} = W^{\text{e}}_{\text{out},1}, \qquad\qquad (23\text{b})$$

$$w^{\text{e}}_{\text{out},i} = \frac{W^{\text{e}}_{\text{in},i} - F^{\text{e}}_i + 2P^{\text{e}}_i}{S^{\text{e}}_i} + 1, \qquad\qquad \forall i \in [1,2], \quad (23\text{c})$$

$$W^{\text{e}}_{\text{out},i} = b^{\text{e}}_{\text{conv},i} w^{\text{e}}_{\text{out},i} + (1 - b^{\text{e}}_{\text{conv},i}) W^{\text{e}}_{\text{in},i}, \qquad\qquad \forall i \in [1,2], \quad (23\text{d})$$

$$W^{\text{e}}_{\text{out},2} \geq 1 \qquad\qquad (23\text{e})$$

Eq. (23a) and Eq. (23b) define the input sizes as the MNIST image input size $W^{\text{e}}_{\text{in},1} = 28$ for the first layer and the output size of the previous convolution for ensuing layers. Eq. (23c) defines the layer $i$ output $w^{\text{e}}_{\text{out},i}$ given the filter size $F^{\text{e}}_i$, padding $P^{\text{e}}_i$, and stride $S^{\text{e}}_i$. Eq. (23d) ensures that the actual convolutional layer output $W^{\text{e}}_{\text{out},i}$ only takes the value of $w^{\text{e}}_{\text{out},i}$ if the layer is active. Finally, Eq. (23e) enforces the output size of the encoder to be at least one.

We add similar auxiliary variables and constraints for the deconvolutional layers:

$$W^{\text{d}}_{\text{in},i} \in \mathbb{N}_0, \qquad\qquad \forall i \in [1,2], \quad (24\text{a})$$

$$w^{\text{d}}_{\text{out},i} \in \mathbb{N}_0, \qquad\qquad \forall i \in [1,2], \quad (24\text{b})$$

$$W^{\text{d}}_{\text{out},i} \in \mathbb{N}_0, \qquad\qquad \forall i \in [1,2], \quad (24\text{c})$$

$$b^{\text{d}}_{\text{dec},i} \in \{0,1\}, \forall i \in [1,2], \qquad\qquad (24\text{d})$$

$$N^{\text{d}}_{\text{dec}} = b^{\text{e}}_{\text{dec},1} + b^{\text{e}}_{\text{dec},2}, \qquad\qquad (24\text{e})$$

$$b^{\text{d}}_{\text{dec},1} \geq b^{\text{e}}_{\text{dec},2}, \qquad\qquad (24\text{f})$$

$$b^{\text{d}}_{\text{dec},1} \rightarrow W^{\text{d}}_{\text{out},2} = 28, \qquad\qquad (24\text{g})$$

$$\neg b^{\text{d}}_{\text{dec},1} \rightarrow C^{\text{d}}_1 = 16 \qquad\qquad (24\text{h})$$

The Eq. (24g) indicator constraint restricts the decoder output to be the original image size $W^{\text{d}}_{\text{out},2} = 28$ if deconvolutional layers are active. Another indicator constraint Eq. (24h) handles the aforementioned case where no deconvolutional layer is active and $C^{\text{d}}_1 = 16$ ensures that the decoder output size can be resized to original MNIST image size. We emphasize that this rule is introduced to simplify the feasible architecture search for methods that do not support explicit input constraints. Note that LEAF-GP could add additional constraints to ensure the architecture's output size can always be resized to the original image size of $28 \times 28$.

$$W^{\text{d}}_{\text{in},1} = 7, \qquad\qquad (25\text{a})$$

$$W^{\text{d}}_{\text{in},2} = W^{\text{d}}_{\text{out},1}, \qquad\qquad (25\text{b})$$

$$w^{\text{d}}_{\text{out},i} = S^{\text{d}}_i(W^{\text{d}}_{\text{in},i} - 1) + F^{\text{d}}_i - 2P^{\text{d}}_i + O^{\text{d}}_i, \qquad\qquad \forall i \in [1,2], \quad (25\text{c})$$

$$S^{\text{d}}_i \geq O^{\text{d}}_i + 1, \qquad\qquad \forall i \in [1,2], \quad (25\text{d})$$

$$W^{\text{d}}_{\text{out},i} = b^{\text{d}}_{\text{conv},i} w^{\text{d}}_{\text{out},i} + (1 - b^{\text{d}}_{\text{conv},i}) W^{\text{d}}_{\text{in},i}, \qquad\qquad \forall i \in [1,2] \quad (25\text{e})$$

Similar to the encoder, Eq. (25) defines constraints for feasible decoder layers. For deconvolutional layers, we also tune output padding $O^{\text{d}}_i$. According to the PyTorch [24] documentation, output padding must be smaller than either stride or dilation. Given that we do not optimize dilation in

deconvolutional layers, Eq. (25d) enforces output padding to be smaller than stride. We introduce similar constraints for fully-connected layers in both the encoder and decoder:

$$b_{\text{fc},i} \in \{0,1\}, \qquad\qquad\qquad \forall i \in [1,4]\,, \quad (26\text{a})$$

$$N_{\text{fc}} = b_{\text{fc},1} + b_{\text{fc},2}, \qquad\qquad\qquad (26\text{b})$$

$$N_{\text{fc}}^{\text{d}} = b_{\text{fc},3} + b_{\text{fc},4}, \qquad\qquad\qquad (26\text{c})$$

$$b_{\text{fc},1} \geq b_{\text{fc},2}, \qquad\qquad\qquad (26\text{d})$$

$$b_{\text{fc},3} \geq b_{\text{fc},4} \qquad\qquad\qquad (26\text{e})$$

To break symmetries in the benchmark problem, we add constraints (27a)–(27m):

$$\neg b_{\text{conv},i}^{\text{e}} \rightarrow C_i^{\text{e}} \leq 4, \qquad\qquad \forall i \in [1,2]\,, \quad (27\text{a})$$

$$\neg b_{\text{conv},i}^{\text{e}} \rightarrow S_i^{\text{e}} \leq 1, \qquad\qquad \forall i \in [1,2]\,, \quad (27\text{b})$$

$$\neg b_{\text{conv},i}^{\text{e}} \rightarrow P_i^{\text{e}} \leq 0, \qquad\qquad \forall i \in [1,2]\,, \quad (27\text{c})$$

$$\neg b_{\text{conv},i}^{\text{e}} \rightarrow F_i^{\text{e}} = 2, \qquad\qquad \forall i \in [1,2]\,, \quad (27\text{d})$$

$$\neg b_{\text{conv},i}^{\text{e}} \rightarrow Act_i^{\text{e}} = \text{ReLU}, \qquad\qquad \forall i \in [1,2]\,, \quad (27\text{e})$$

$$\neg b_{\text{fc},i} \rightarrow Act_i^{\text{fc}} = \text{ReLU}, \qquad\qquad \forall i \in [1,4]\,, \quad (27\text{f})$$

$$\neg b_{\text{fc},1} \rightarrow N_1^{\text{fc}} \leq 0, \qquad\qquad (27\text{g})$$

$$\neg b_{\text{dec},i}^{\text{d}} \rightarrow C_i^{\text{d}} \leq 4, \qquad\qquad \forall i \in [1,2]\,, \quad (27\text{h})$$

$$\neg b_{\text{dec},i}^{\text{d}} \rightarrow S_i^{\text{d}} \leq 1, \qquad\qquad \forall i \in [1,2]\,, \quad (27\text{i})$$

$$\neg b_{\text{dec},i}^{\text{d}} \rightarrow P_i^{\text{d}} \leq 0, \qquad\qquad \forall i \in [1,2]\,, \quad (27\text{j})$$

$$\neg b_{\text{dec},i}^{\text{d}} \rightarrow O_i^{\text{d}} \leq 0, \qquad\qquad \forall i \in [1,2]\,, \quad (27\text{k})$$

$$\neg b_{\text{dec},i}^{\text{d}} \rightarrow F_i^{\text{d}} = 2, \qquad\qquad \forall i \in [1,2]\,, \quad (27\text{l})$$

$$\neg b_{\text{dec},1}^{\text{d}} \rightarrow Act_1^{\text{d}} = \text{ReLU} \qquad\qquad (27\text{m})$$

Constraints (27a)–(27m) set layer-specific hyperparameters to pre-defined default values when the associated layer is inactive. We select these defaults as the lower bound for non-categorical variables and the first category for categorical variables. While SMAC is unable to handle more complicated constraints restricting outputs of deconvolutional layers, it can handle hierarchical search space structures. For VAE-NAS benchmark runs using SMAC as an optimizer we enforce hierarchies according to constraints (27a)–(27m) which deactivate hyperparameters for inactive layers.

Table 4: Hyperparameter names, types, and domains for the VAE-NAS benchmark. The transformation column refers to post-processing computations before passing the hyperparameter value to the neural network training. The architecture with all layers activated comprises `C1-C2-FC1-FC2-L-FC3-FC4-D1-D2`, with L referring to the latent space layer.

| # | Name | Type | Domain | Transformation |
|---|------|------|--------|----------------|
| | **General** | | | |
| 0 | Learning rate | conti. | $[-4.0, -2.0]$ | $\alpha = 10^{x_0}$ |
| 1 | Latent space size | integer | $[16, 64]$ | $N^{\text{lat}} = x_1$ |
| 2 | Num. conv. enc. layers | integer | $[0, 2]$ | $N^{\text{e}}_{\text{conv}} = x_2$ |
| 3 | Num. fully-conn. enc. layers | integer | $[0, 2]$ | $N^{\text{e}}_{\text{fc}} = x_3$ |
| 4 | Num. deconv. dec. layers | integer | $[0, 2]$ | $N^{\text{d}}_{\text{dec}} = x_4$ |
| 5 | Num. fully-conn. dec. layers | integer | $[0, 2]$ | $N^{\text{d}}_{\text{fc}} = x_5$ |
| | **Encoder** | | | |
| | **Convolutional layer 1** (`C1`) | | | |
| 6 | Number of output channels | integer | $[2, 5]$ | $C^{\text{e}}_1 = 2^{x_6}$ |
| 7 | Stride | integer | $[1, 2]$ | $S^{\text{e}}_1 = x_7$ |
| 8 | Padding | integer | $[0, 3]$ | $P^{\text{e}}_1 = x_8$ |
| 9 | Filter size | categ. | $\{3, 5\}$ | $F^{\text{e}}_1 = x_9$ |
| 10 | Activation function | categ. | $\{\text{ReLU}, \text{PReLU}, \text{Leaky ReLU}\}$ | $Act^{\text{e}}_1 = x_{10}$ |
| | **Convolutional layer 2** (`C2`) | | | |
| 11 | Number of output channels | integer | $[3, 6]$ | $C^{\text{e}}_2 = 2^{x_{11}}$ |
| 12 | Stride | integer | $[1, 2]$ | $S^{\text{e}}_2 = x_{12}$ |
| 13 | Padding | integer | $[0, 3]$ | $P^{\text{e}}_2 = x_{13}$ |
| 14 | Filter size | categ. | $\{3, 5\}$ | $F^{\text{e}}_2 = x_{14}$ |
| 15 | Activation function | categ. | $\{\text{ReLU}, \text{PReLU}, \text{Leaky ReLU}\}$ | $Act^{\text{e}}_2 = x_{15}$ |
| | **Fully-connected layer 1** (`FC1`) | | | |
| 16 | Number of nodes | integer | $[0, 15]$ | $N^{\text{fc}}_1 = 64 \times x_{16}$ |
| 17 | Activation function | categ. | $\{\text{ReLU}, \text{PReLU}, \text{Leaky ReLU}\}$ | $Act^{\text{fc}}_1 = x_{17}$ |
| | **Fully-connected layer 2** (`FC2`) | | | |
| 18 | Activation function | categ. | $\{\text{ReLU}, \text{PReLU}, \text{Leaky ReLU}\}$ | $Act^{\text{fc}}_2 = x_{18}$ |
| | **Decoder** | | | |
| | **Fully-connected layer 3** (`FC3`) | | | |
| 19 | Activation function | categ. | $\{\text{ReLU}, \text{PReLU}, \text{Leaky ReLU}\}$ | $Act^{\text{fc}}_3 = x_{19}$ |
| | **Fully-connected layer 4** (`FC4`) | | | |
| 20 | Activation function | categ. | $\{\text{ReLU}, \text{PReLU}, \text{Leaky ReLU}\}$ | $Act^{\text{fc}}_4 = x_{20}$ |
| | **Deconvolutional layer 1** (`D1`) | | | |
| 21 | Number of input channels | integer | $[3, 6]$ | $C^{\text{d}}_1 = 2^{x_{21}}$ |
| 22 | Stride | integer | $[1, 2]$ | $S^{\text{d}}_1 = x_{22}$ |
| 23 | Padding | integer | $[0, 3]$ | $P^{\text{d}}_1 = x_{23}$ |
| 24 | Output Padding | integer | $[0, 1]$ | $O^{\text{d}}_1 = x_{24}$ |
| 25 | Filter size | categ. | $\{3, 5\}$ | $F^{\text{d}}_1 = x_{25}$ |
| 26 | Activation function | categ. | $\{\text{ReLU}, \text{PReLU}, \text{Leaky ReLU}\}$ | $Act^{\text{d}}_1 = x_{26}$ |
| | **Deconvolutional layer 2** (`D2`) | | | |
| 27 | Number of input channels | integer | $[2, 5]$ | $C^{\text{d}}_2 = 2^{x_{27}}$ |
| 28 | Stride | integer | $[1, 2]$ | $S^{\text{d}}_2 = x_{28}$ |
| 29 | Padding | integer | $[0, 3]$ | $P^{\text{d}}_2 = x_{29}$ |
| 30 | Output Padding | integer | $[0, 1]$ | $O^{\text{d}}_2 = x_{30}$ |
| 31 | Filter size | categ. | $\{3, 5\}$ | $F^{\text{d}}_2 = x_{31}$ |