# OpenReview forum: "Tree ensemble kernels for Bayesian optimization with known constraints over  mixed-feature spaces"
_NeurIPS.cc/2022/Conference — NeurIPS 2022 Accept_

### Official Review · Reviewer_PeTE · 2022-07-08

**Rating:** 8
**Confidence:** 4
**Soundness:** 4 excellent
**Presentation:** 4 excellent
**Contribution:** 4 excellent

**Summary:**

This paper proposes a feasible way to perform Bayesian optimization using ensembles of decision trees. Although it is easy to use decision trees ensembles to predict a mean and variance (and therefore formulate acquisition functions like upper confidence bounds), _optimizing_ these acquisition functions is challenging because they are piecewise constant. The key insight of this paper is that if the decision tree predictions are used to form a GP kernel, upper confidence bound acquisition function can be formulated as a conic problem and be inputted into a generic solver to find the global optimum. A secondary advantage of this approach is that the solver can also tolerate a large number of constraints from a variety of families, which is practically useful.

The authors provide the technical details of this formulation and show that it works well for a variety of synthetic optimization problem and one real-world problem over mixed-variable, hierarchical spaces.

**Questions:**

- Trees are effectively parametric methods: once trained on data, updating the trees generally requires retraining. This is in contrast to the GP with the tree kernel, which will make different predictions conditioned on new data even if the underlying trees are not retrained. _In your baseline experiments, do you keep the trees fixed or do you retrain them?_ If you are not retraining them, this could explain the roughly constant performance of the baselines in e.g. Figure 4. I think that you should be retraining them.
- _Do all tree-based methods use the same underlying trees?_ To minimize external sources of variation, I think that the trees in SMAC/skopt should be the exact same trees used in the GP kernel. If this is not possible/extremely hard to do then don't worry about it.
- _What things are randomized in the random seeds?_ In general I think the main 2 sources of randomness are 1) randomness in the BO procedure (e.g. initialization of interior point methods) and 2) randomness in training of the tree ensembles. Do your error bars in Figures 2-4 cover both of these sources of variation, or just the randomness of BO?
- _How necessary is the gurobi solver specifically?_ Could you use another similar solver? Would other solvers be able to handle the same constraints?
- The solver seems to be able to handle a large family of constraints. _Is the solution exact/nearly exact for all these constraint types, or does the solver lose its guarantees for certain constraint families (e.g. non-convex constraints)?_ (I'm not very familiar with cone solvers)

**Limitations:**

As far as I can tell, the limitations of this work are discussed adequately.

**Strengths And Weaknesses:**

Overall I thought that this paper was very good.

### Strengths

- **Very relevant problem choice**: trees are one of the few models that can easily handle mixed-variable/hierarchical data, _and_ work well on small datasets. To my knowledge, optimizing the acquisition function is a significant hurdle preventing the widespread use of tree ensembles in Bayesopt.
- **Elegant solution**: formulating the acquisition function as a cone problem allows for decades of research on convex solvers to be applied. This is a clever insight. It is nice to see a (semi-)exact solution to an optimization problem presented at NeurIPS, where most papers tend to use approximations which are hard to verify.
- **Extremely well-written**: this paper is definitely in the top 10% of writing quality. Everything was extremely clear. I especially appreciated the discussion of the related work and the technical background (section 3). The technical details in section 4 were very well presented. I don't know if I would 100% be able to implement the method based on the paper (the explanation of $\nu$ and $v$ in equations 3-4 could be slightly clearer), but it is closer to this ideal than perhaps 90% of papers at NeurIPS. Also, unlike many ML conference papers, it did not feel like the authors crammed a 12 page paper into 9 pages by moving important explanations/equations into the appendix, which I really appreciated.
- **Highly Usable**: this method seems like it should work "out of the box" (assuming access to a conic solver) because it is simple and could be plugged into any decision tree package. I highlight this because I feel like most methods presented at generally do not work "out of the box" because they are brittle, require a lot of tuning, or have undisclosed limitations.

### Weaknesses

- **Several technical details unclear**: see "Questions" section. These are not absolutely critical to the paper but I would appreciate clarification.
- **Unclear effect of "different objective" vs "exact solution"**: the proposed method has 2 deviations from using the empirical mean/variance of a tree ensemble for Bayesopt: 1) making predictions with a tree kernel GP instead of empirical mean/variance 2) optimizing UCB exactly using a cone solver. In section 5 it was generally unclear to me how much of the difference in final performance came from using the trees to make predictions in a different way (GP instead of empirical mean/variance), and how much came from performing [nearly] exact maximization of the acquisition function. A good experiment to do would be use the GP formulation to find a promising set of candidate points (perhaps using multiple seeds), then use the original empirical mean/variance UCB objective to select a point to query in the end. This would be combining "exact optimization" with the standard way of making predictions.
- **Code requires a non-public library** gurobi. This is not a deal-breaker, and I realize there are academic licenses available, but it would be really nice to have a fully open-source version for the camera-ready paper. I ask some additional questions about the choice of solver below.

### Suggestions

- Given that the paper focuses specifically on non-continuous spaces, why is $x$ defined to be a real number when introducing Bayesopt on line 44? I think $x$ should just be an element of a general space $\mathcal{X}$
- Many readers at NeurIPS may not be very familiar with cone problems / conic solvers. It would be nice to put some information on these in the background, even if it is just a short section saying that "it is a very general way of formulating certain optimization problems, with a variety of established methods for solving them"
- "LCB" is a fairly uncommon formulation of Bayesopt; I think many readers are accustomed to seeing problems posed as UCB _maximization_. Perhaps it would be slightly helpful to the reader to frame the problem as maximization and multiple everything in the paper by $-1$?
- I saw in the appendix that you use Adam to optimize GP kernel hyperparameters. One problem with Adam is that it may converge very slowly if the learning rate is too low, or not fully converge if it is too large. Especially since you are only optimizing 2 hyperparameters, I recommend using L-BFGS instead. In my personal experience this method is much more robust.
- The objective in equation 12 would also be another interesting baseline to see: it would show how much the GP variance actually contributes, as opposed to just finding a point with a sufficient amount of disagreement among ensemble members.


### Overall

Originality: seems high, but I am not very familiar with related work in this area

Quality: high, good paper, good method, good evaluation

Clarity: very high!

Significance: high. The proposed method solves a real problem and is likely to be used in practice, especially if a good open-source library is published.

I definitely recommend accepting this paper and my score reflects that. I would consider increasing my score slightly if my suggestions are addressed. I could also imagine decreasing my score if:
- The novelty is not as high as I thought (I am not very familiar with related work in this area, so perhaps there is a paper that does something very similar which was not cited)
- There were technical flaws that I did not catch (I know only the basics of convex optimization/cone problems, so this is possible)

---

> ### Author Response · Authors · 2022-08-02
> **Official Response to Comments of Reviewer PeTE**
>
> Thanks for the encouragement and also for the points of constructive criticism that
>     will help us improve.
> We are indeed excited about the connection between the acquisition function and
>     conic optimization.
>
> **Strength 3 [Variables $\boldsymbol{\nu}$ and parameters $\boldsymbol{v}$]** It's a good point that
>     these definitions could be clearer.
> Our response to Reviewer 94qs (Minor Comment 1) documents how we will improve these definitions.
>
> **Weakness 2 [Unclear effects of *different objective* vs. *exact solution*]** It's a good point that
>     it's nice to understand what is the effect of each deviation from traditional Bayesian optimization.
> Beyond *different objective* vs. *exact solution*, there is one more confounding factor:
>     *feasibility of the input constraints*.
> For the Sections 5.3 and 5.4 test instances where LEAF-GP is performing best, it's this
>     feasibility test that's most important.
> Note in Sections 5.3 and 5.4 how well FEAS-RND performs: the difference between
>     FEAS-RND and LEAF-GP is the result of the two confounding factors *different objective* and
>     *exact solution*.
>
> Section 5.2 isolates the contribution of the *exact solution*. In these graphs, we don't claim
>     that LEAF-GP outperforms standard Bayesian optimization: we only claim that LEAF-GP is performing
>     roughly *with* the state-of-the-art in Bayesian optimization for unconstrained problems.
> Section 5.2 does, however, show that using global optimization does improve the result, e.g.,
>     compare LEAF-GP (using global optimization) to LEAF-GP-RND (not using global optimization).
>
> **Weakness 3 \& Question 4 [Non-public library Gurobi]** Open-source optimization solvers with similar
>     algorithms include: Bonmin (EPL), MindtPy (BSD), and SHOT (EPL).
> As the reviewer probably knows, commercial optimizers like Gurobi tend to solve larger
>     optimization problems more quickly, so we don't know if the open-source solvers would be
>     effective on the largest instances.
> However, the open-source solvers do work equivalently from an algorithmic point-of-view, and we will
>     mention them in the paper.
>
> **Suggestion 1 [Line 44]** Thanks for catching the typo. We agree that $\boldsymbol{x}$ should just
>     be an element of a general space $\boldsymbol{X}$.
>
> **Suggestion 2 [Conic optimization]** It's a great idea to add background information on mixed-integer
>     conic optimization since this is an area that has really jumped forward in recent years.
> We will add a line or two to Section 1 (Introduction).
>
> **Suggestion 3 [LCB to UCB]** Good point. We will switch the sense of our equations to *maximization*.
>     This will also switch the directionality of our graphs.
>
> **Suggestion 4 [ADAM vs L-BFGS]** Thanks for providing these highly relevant insights.
> We will aim to provide an option for hyperparameter tuning via L-BFGS when releasing the
>     accompanying code and/or open-source tool.
>
> **Suggestion 5 [Different objective?]** We agree that this would be an interesting baseline, and
>     would be an interesting area for future study.
>
> **Question 1 [Retraining trees]** Yes, we retrain the trees at every iteration.
>     We will clarify this in Section 4.
>
> **Question 2 [Do methods use the same underlying trees?]** Each different tree-based method uses its
>     own approach for training trees, so the resulting trees are typically different.
> It's true that it would be difficult to use the same trees in every method.
> Additionally, each piece of software such as SMAC and SKOPT has its own hyperparameters:
>     engineers have thought carefully about how to put the different algorithmic
>     components together.
> We don't want to force every approach to use the same trees because this change would almost
>     certainly weaken SKOPT and SMAC: the codes have been stress tested using different ways of
>     generating trees.
>
> **Question 3 [What is randomized?]** We're randomizing the five initial points.
> Because those five initial points are randomized, the tree training and the hyperparameter tuning
>     will change from run to run.
>
> **Question 5 [Large family of constraints?]** We use Gurobi to solve a mixed-integer optimization
>     problem with both second-order conic and linear constraints.
> So the constraints can be linear or second-order conic.
> Additionally, we can use discrete and/or continuous variables.
> Because of the advances of Hijazi et al. [2014] and Vielma et al. [2017] in developing extended
>     formulations that lift second-order conic constraints into a higher dimension, optimization
>     of mixed-integer second-order cone programs is now (practically) as well-developed as
>     optimization of mixed-integer linear programs.

---

### Official Review · Reviewer_94qs · 2022-07-11

**Rating:** 6
**Confidence:** 3
**Soundness:** 3 good
**Presentation:** 2 fair
**Contribution:** 3 good

**Summary:**

The paper develops a new framework for Bayesian optimization with mixed-type features and flexible feature constraints. The idea is to utilize a GP model with a kernel function based on tree ensembles that can naturally handle mixed-type features. A mixed-integer optimization technique is used to minimize the piecewise-constant LCB acquisition function. The proposed method shows promising performance in various benchmarks.

**Questions:**

Major comments:

1. As far as I understand, there are two possible ways to train the proposed surrogate model with a tree ensemble kernel: (i) treat the tree structure as a kernel hyperparameter and learn it by maximizing marginal likelihood (which is an end-to-end approach and provides a coherent statistical model), and (ii) fit the tree structure first by gradient boosting, and then fit a GP model with the given tree structure (which does not seem to be a well-defined statistical model to me). It is unclear to me which approach is used. If (ii) is used, what are the responses used to train boosting trees?

2. The experiment in Section 5.1 aims to verify the uncertainty metrics. However, in Eq. (12a) only the posterior mean is included while the posterior variance (which accounts for uncertainty) is missing. Moreover, the plots in Figure 2 only show the error for the mean prediction, which again does not convey any information on uncertainty. It is unclear to me how the conclusion in Section 5.1 is reached and why the constraint in Eq. (12c) is needed.

3. Since the posterior mean and variance from the tree-kernel GP are piecewise constant functions corresponding to the tree partitions, I wonder how sensitive the performance is with respect to the tree ensemble hyperparameters such as the number of boosting iterations.

4. How is the proposed method compared to its competitors in terms of computation time? I would expect the sparsity in the tree-based covariance matrix can reduce computation for GP.

Minor comments:

* I think the notations in Section 3.3 are quite confusing and should be clearly defined. For example, what do $v_{i, j}$, $\nu_{i, j}$ and $\mathrm{splits}(t)$ mean?
* In Eq. (5a), I understand that the optimization is done over $\mathbf{x}$. But since $\mathbf{z}$ is a function of $\mathbf{x}$ given the tree structures (as in Section 3.1), it should not be included under $\mathrm{argmin}$. It is also not clear what $\nu$ is.
* Line 98: Leaf subspace $\mathbf{x}_l \in \mathbb{R}^n$ -> $\mathbf{x}_l \subset \mathbb{R}^n$.


**Limitations:**

Please see the major comments above.

**Strengths And Weaknesses:**

Strength:

* A novel framework for Bayesian optimization to handle mixed-type features and flexible feature constraints.
* Promising performance in various benchmarks.

Weakness (see major comments below for details):

* Training of the surrogate model is unclear.
* The experiment in Section 5.1 does not seem solid.
* The paper could benefit from additional sensitivity and timing results.

---

> ### Author Response · Authors · 2022-08-02
> **Official Response to Comments of Reviewer 94qs**
>
> Thanks for this very useful review that will improve our paper.
> This response describes the experimental setup and gives additional computational details.
>
> **Question 1 [Training the tree ensemble]** We take the second approach by first fitting
>     the tree structure using the output $f$ as the response and then fitting a GP model with the
>     given tree structure.
> We agree that jointly training the kernel parameters would be elegant, but the idea
>     would be difficult to implement because, for example, we couldn't use
>     existing tree-ensemble libraries.
> The main contribution of our paper is the optimization approach, but we agree that the joint
>     training could really improve things in future work.
>
>
> We will clarify our approach in Section 3.1.
>
> **Question 2a [Verifying uncertainty metrics]** The posterior variance is omitted
>     from the objective (Eq. 12a) because we added the vector of posterior covariances
>     to a constraint (Eq. 12c).
> Eq. (12c) is equivalent to Eq. (8b) and therefore restricts the largest element of
>     Eq. (8a) to be less than or equal to $R$.
>
> We moved the posterior variance to a constraint to study how both the objective and the
>     model error change with the tree agreement ratio.
> The tree agreement ratio, given in Eq. (8b), encodes the covariance of $\boldsymbol{x}$ with
>     individual data points $\boldsymbol{x}_i$.
> Equivalently, we could formulate Eqs. (12a) - (12c) as a multiobjective problem where the two
>     objectives are (i) posterior mean and (ii) the largest possible value for Eq.\(8b).
> But we think Eq. (13) and Figure 2b would be less clear with a multiobjective formulation.
>
> We will clarify the optimization formulation in Section 5.1. We will also clarify in the Figure 2
>     caption that changing the maximum tree agreement ratio is equivalent to changing the
>     maximum posterior covariance.
>
> **Question 3 [Sensitivity to tree hyperparameters]** Unfortunately, we don't have a
>     sensitivity study showing performance with respect to the tree hyperparameters.
> To mitigate the issue, Appendix B.1 documents how the hyperparameters are constant across
>     all problem instances.
> The only exceptions are the CIFAR-NAS benchmark and the new variational autoencoder example:
>     these problems are large and we increased the maximum depth per decision tree (to 5 for both problems)
>     and the total number of trees (to 100 for both problems).
>
> **Question 4a [Computation time]** In our implementation, the average performance of
>     LEAF-GP is up to 10 times slower than other approaches.
> We can't give a more precise assessment of computation time because computations were conducted on
>     an HTCondor cluster with machines of similar specifications.
>
> We are fine with LEAF-GP being relatively slow because we're interested in the Bayesian
>     optimization setting with significant time to chose a next evaluation point.
> However, if we did want to speed up LEAF-GP, we would first limit the time we give Gurobi
>     (currently set to 100 s).
> This would maybe mean that we were no longer getting the exact solution to the mixed-integer
>     second-order conic optimization problem, but we would at least (probably) do better than
>     the FEAS-RND line in Figures 4 and 5 and would obtain lower and upper bounds to assess how good the
>     solution already is.
>
> **Question 4b [Sparsity in the tree-based covariance matrix]** Good point! We already mention
>     Lee et al. [CVPR 2015] which analyses the complexity improvement based on the sparsity
>     in the tree-based covariance matrix.
> We will add 1-2 sentences to highlight the connection to this paper.
>
> **Minor Comment 1 [Notations]** We kept Section 3.3 short because there's nothing new: these are
>     previous contributions of Mišić [2020] and Mistry et al. [2021].
> However, it's a good point that, from the Section 3.3, it should at least be clear what are
>     the arising equations.
> We will clarify:
> - Parameters $\boldsymbol{v}$ are the splitting thresholds learned by the gradient boosting.
>   These parameters are uniquely determined by the trained ensemble.
> - Binary variables $\boldsymbol{\nu}$ activate exactly one interval for each continuous feature.
>   Mišić [2020] describes why variables $\boldsymbol{\nu}$ are different than variables
>   $\boldsymbol{z}$: they enforce consistent input domains between trees.
> - The *split* directions *left* and *right* encode the left/right directions
>   (with respect to the root node) to get to a leaf of a specific tree.
>   These parameters are uniquely determined by the trained ensemble.
>
> **Minor Comment 2 [Decision variables]** We agree it would be more clear to remove the auxiliary
>     variables $\boldsymbol{z}$ and $\boldsymbol{\nu}$ from Eq. (5a).
>
> **Minor Comment 3 [Leaf subspace]** Thanks for noticing the typo, we will correct it.

---

### Official Review · Reviewer_NYXt · 2022-07-13

**Rating:** 5
**Confidence:** 3
**Soundness:** 3 good
**Presentation:** 4 excellent
**Contribution:** 2 fair

**Summary:**

This paper focuses on Bayesian Optimization and utilizes Gaussian Processes (GPs) with ensemble tree kernels. The proposed approach advocates a mixed-integer second-order cone optimization formulation, relying on the lower confidence bound (LCB) acquisition criterion, that allows for uncertainty quantification and readily accommodates the incorporation of input constraints. Numerical tests show the benefits of the proposed framework compared to existing methods in the literature.

**Questions:**

Some minor comments are listed below:
- In Eq. 6b, instead of $\sigma^2$ it should be $\sigma^2(\mathbf{x})$.
- In line 51, instead of “prior work” it should be “prior works”.
- In line 118, instead of “where data is sparse” it should be “where data are sparse”.
- In line 167, instead of “We compute constant matrix”, it should be “We compute the constant matrix”.
- In line 199, in the phrase “we found that Gurobi 9 often solves” a citation is needed.


**Limitations:**

The limitations subsection is well written. Some additional concerns of mine are mentioned in the “Strengths and Weaknesses” section of the review.


**Strengths And Weaknesses:**

STRENGTHS

- The ensemble tree kernel GP - based approach is intuitive and seems to work well in practice.
- The paper is well written and the ideas of the paper are well presented.

WEAKNESSES

- The novelty of the paper is somewhat limited. The ensemble tree kernel is known and the same holds true for Eqs. 3 and 4. In my mind, the advocated approach is just a combination of some intuitive well-known components.
-  Additional experiments are needed to corroborate the benefits of the proposed approach compared to existing methods in the literature. Except the GP-UCB approach which is indeed a good baseline, there exist many other well-known acquisition criteria combined with GP surrogate model such as Expected Improvement (EI), Thompson Sampling (TS), Entropy Search (ES) and Predictive Entropy Search (PES) to list a few. So, in my opinion it is important to compare with some of these approaches to better demonstrate the benefits of the advocated method, and especially with the EI which seems to be very effective in several BO problems. In addition, there exist some very interesting ensemble approaches that can effectively handle continuous-categorical data; see e.g [1], [2]. A numerical comparison (or even discussion) with these approaches would also be nice.

[1] Gopakumar, Shivapratap, Sunil Gupta, Santu Rana, Vu Nguyen, and Svetha Venkatesh. "Algorithmic assurance: An active approach to algorithmic testing using bayesian optimisation." Advances in Neural Information Processing Systems 31 (2018).

[2] Nguyen, Dang, Sunil Gupta, Santu Rana, Alistair Shilton, and Svetha Venkatesh. "Bayesian optimization for categorical and category-specific continuous inputs." In Proceedings of the AAAI Conference on Artificial Intelligence, vol. 34, no. 04, pp. 5256-5263  (2020).

---

> ### Comment · Reviewer_PeTE · 2022-07-27
> **Clarification of what the additional requested baselines would provide for the paper**
>
> In your list of weaknesses you state:
>
> > Additional experiments are needed to corroborate the benefits of the proposed approach compared to existing methods in the literature. Except the GP-UCB approach which is indeed a good baseline, there exist many other well-known acquisition criteria combined with GP surrogate model such as Expected Improvement (EI), Thompson Sampling (TS), Entropy Search (ES) and Predictive Entropy Search (PES) to list a few. So, in my opinion it is important to compare with some of these approaches to better demonstrate the benefits of the advocated method, and especially with the EI which seems to be very effective in several BO problems.
>
> To be clear, are you asking the authors to use the Matern kernel GP with these acquisition functions? If so, I don't think that these experiments would provide much value to the paper. If the results from LEAF-GP are better, that is good. If they are worse, then this is just an example of a task where the Matern GP fits the data well. There are surely other cases where the Matern GP will fit the data less well (e.g. using categorical variables), in which case there is little hope for good BO performance. Put another way, performing _worse_ than these extra baselines in no way disproves or refutes the authors' claim that this method is useful. I don't want to ask the authors to perform additional experiments if the outcome does not impact the accuracy of the paper's key claims...
>
> More broadly, I don't think that the acquisition function is very important compared to the model fit: in my experience, if a model is highly misspecified, then BO with any acquisition function will perform poorly. If the model fit is good, then it is hard to predict which acquisition function will perform better (it's basically random).

---

> ### Author Response · Authors · 2022-08-02
> **Official Response to Comments of Reviewer NYXt**
>
> We appreciate these thoughtful comments. This response clarifies our contributions and describes
>     how we will better position our paper with respect to prior contributions.
>
> **Weakness 1 [Novelty]** Agreed that the tree kernel (Section 3.1) and Eqs. (3) and (4) are not new.
> Indeed, *none* of Section 3 is new: Section 3 introduces the prior work that we use to make
>     the contributions in Section 4.
> Our contribution (described in Section 4) is to develop a way of optimizing the acquisition function
>     created by combining the already-known elements in Section 3.
>
> The reason for this misunderstanding is an unfortunate typo on Line 36.
> We will update that line to read *Section 4* instead of Section 3.
> We will additionally add a sentence in Section 1 to clarify that Section 3 reviews prior work.
>
> **Weakness 2a [Additional baseline comparison]** In the updated paper, we now include
>     Expected Improvement (EI) with the Matérn kernel.
> As expected from the literature, EI performs well on the Section 5.2 unconstrained benchmarks.
> Nevertheless, EI does not perform well for the constrained problems, where we need to
>     penalize the objective rather than add constraints.
> This highly-constrained setting is exactly where we expect LEAF-GP to perform well.
>
> **Weakness 2b [Additional literature]** Thanks for these nice papers.
> Gopakumar et al. [2018] and Nguyen et al. [2020] develop good approaches for optimizing
>     over mixed-feature spaces.
> We will add the papers to the Section 2 (Related Work) discussion.
> Our claims of novelty remain because, unlike these papers, we also allow input constraints.
> We would have the same difficulty numerically comparing with these works, as we do in the other
>     comparisons in Sections 5.3 and 5.4: we would need to artificially add some of the input
>     constraints by penalizing the objective since the approach of Gopakumar et al. [2018] and
>     Nguyen et al. [2020] does not immediately admit input constraints.
>
> **Questions [5 minor comments]** We appreciate the reviewer spotting these issues and will
>     make all the proposed changes.

---

> > ### Comment · Reviewer_NYXt · 2022-08-09
> > **Response to the Authors**
> >
> > I would like to thank the authors for their response and I appreciate their time and effort for addressing my comments. I increase my score to Borderline Accept.

---

### Author Response · Authors · 2022-08-02
**Official Comment regarding Rebuttal Revision**

Sincere thanks to the reviewers for their constructive, thoughtful comments.

After submitting, we realized we had missed a SMAC feature that improves SMAC's performance on
    our Section 5.4 CIFAR-NAS example.
The feature ([link](https://automl.github.io/ConfigSpace/master/API-Doc.html#conditions)) integrates
    hierarchical constraints into SMAC.
Using this feature, SMAC now performs slightly better than our LEAF-GP approach for
    CIFAR-NAS. Only the SMAC line in Figure 5b is affected: none of the other examples are affected
    by this change. Our revised paper has an updated Figure 5b.

Further experimentation identified other neural architecture search examples where LEAF-GP
    outperforms SMAC, for example the Daxberger et al. [IJCAI 2021] variational autoencoder example.
LEAF-GP outperforms SMAC on highly constrained neural architecture search examples
    where LEAF-GP can represent constraints that SMAC can't.
Because LEAF-GP can handle *any* input constraint represented by mixed-integer
    optimization constrained by second-order cones, LEAF-GP can take a wider array of
    input constraints than SMAC.

To clarify these changes, we have taken the following actions in the revised paper:

- We updated Figure 5b so that SMAC appears to its greatest strength.
- We changed the claim on Line 302 to mention that LEAF-GP does not outperform SMAC on this benchmark.
- We mentioned in the paper that we have another NAS benchmark where LEAF-GP performs well against SMAC and
    included that example in the appendix (see: Fig. 9, Section F in Rebuttal Revision).

We have not yet incorporated the (very nice) changes suggested by the reviewers, but the following comments document the changes we will make.

---

### Meta-Review · Area_Chair_2XNo · 2022-08-25

**Recommendation:** Accept
**Confidence:** Certain

**Metareview:**

This paper presents a fairly neat take on combining tree ensembles with GPs by creating a kernel function from the tree ensembles. This not only allows for Bayesian optimization over discrete and mixed feature spaces--inheriting the usual advantage of tree ensembles--but allows for UCB/LCB to be optimized using a mixed integer programming tool.

Based on reviewer questions and responses, there are a few things that surfaced that I do think the paper would benefit from discussion further in the camera ready version. Most particular, while the setting here is certainly reasonable, I do think the paper would be overall better if there was simply just an ablation of optimization performance versus the time allocated to Gurobi.

**Award:**

No

---

### Decision · Program_Chairs · 2022-09-14

Accept